



# Cross-basin differences in the nutrient assimilation characteristics of induced phytoplankton blooms in the subtropical Pacific waters

Fuminori Hashihama[1,2], Hiroaki Saito[3], Taketoshi Kodama[4,5], Saori Yasui-Tamura[1], Jota Kanda[1], Iwao Tanita[4,6], Hiroshi Ogawa[3], E. Malcolm S. Woodward[7], Philip W. Boyd[2], Ken Furuya[4,8]

[1]Department of Ocean Sciences, Tokyo University of Marine Science and Technology, Tokyo 108-8477, Japan
[2]Institute for Marine and Antarctic Studies, University of Tasmania, Hobart TAS 7004, Australia
[3]Atmosphere and Ocean Research Institute, The University of Tokyo, Chiba 277-8564, Japan
[4]Department of Aquatic Bioscience, Graduate School of Agricultural and Life Sciences, The University of Tokyo, Tokyo 113-8657, Japan
[5]Fisheries Resources Institute, Japan Fisheries Research and Education Agency, Kanagawa 236-8648, Japan
[6]Fisheries Technology Institute, Japan Fisheries Research and Education Agency, Okinawa 907-0451, Japan
[7]Plymouth Marine Laboratory, Prospect Place, The Hoe, Plymouth PL1 3DH, UK
[8]Graduate School of Science and Engineering, Soka University, Tokyo 192-8577, Japan

*Correspondence to*: Fuminori Hashihama (f-hashi@kaiyodai.ac.jp)

**Abstract.** To better understand the nutrient assimilation characteristics of subtropical phytoplankton, deep water addition incubation experiments were carried out on surface waters collected at seven stations across the subtropical North and South Pacific Ocean. These deep water additions induced phytoplankton blooms with nutrient drawdown at all stations. The drawdown ratios of dissolved inorganic nitrogen (DIN) to phosphate ($PO_4$) varied from 14.1 to 29.6 at the $PO_4$-replete stations in the central North Pacific (CNP) and eastern South Pacific (ESP). These ratios were similar to the range represented by the canonical Redfield ratio (16) through to typical particulate N:P ratios in the surface subtropical ocean (28). In contrast, lower DIN:$PO_4$ drawdown ratios (8.0-12.9) were observed in induced blooms at the $PO_4$-depleted stations in the western North Pacific (WNP). The DIN:$PO_4$ drawdown ratios in the $PO_4$-replete ESP were associated with eukaryote-dominated blooms, while those in $PO_4$-depleted WNP were associated with eukaryotic and cyanobacterial blooms. The surplus $PO_4$ assimilation, relative to DIN, by phytoplankton in the WNP was not expected based on their typical cellular N:P ratio, and was likely due to the high $PO_4$ uptake capability as induced by low $PO_4$-adapted phytoplankton. The low and high P* ($=PO_4$-DIN/16) regimes geographically corresponded to the low and high DIN:$PO_4$ drawdown ratios in the WNP and the CNP or ESP, respectively. The basin-wide P* distribution in the oligotrophic Pacific surface waters showed a clear regional trend from low in the WNP (<50 nM) to high in the ESP (>100 nM). These results suggest that the subtropical phytoplankton blooms as observed in our experiments could be an important factor controlling P* as well as the commonly recognized dinitrogen fixation and denitrification characteristics.



# 1 Introduction

The surface waters of the subtropical oceans are characterized by strong stratification, low nutrients, and low phytoplankton biomass (Karl, 2002). In this regime, primary production is largely sustained by regenerated production ($f$-ratio: ~0.1, Dugdale and Goering, 1967; Eppley and Peterson, 1979), and driven by small phytoplankton such as the picocyanobacteria *Prochlorococcus* and *Synechococcus* (Waterbury et al., 1979; Chisholm et al., 1988). Despite the persistent oligotrophic regime, phytoplankton blooms with large diatoms and cyanobacteria occur occasionally in the subtropical oceans and have large impacts on new production and export production (Benitez-Nelson et al., 2007; McGillicuddy et al., 2007; Dore et al., 2008; Wilson and Qiu, 2008; Karl et al., 2012; Villareal et al., 2012; Hashihama et al., 2014). The mechanisms that bring about the development of these blooms are not simple, but fundamentally they involve nutrient supply with physical forcing (Wilson et al., 2013; Toyoda and Okamoto, 2017).

The nutrient supply to surface subtropical oceans is important for many aspects of biogeochemical cycling and food-web dynamics as it drives new production and net community production (Sarmiento and Gruber, 2006; Saito, 2019). Seasonal variations in dissolved inorganic carbon and dissolved oxygen in the subtropical oceans highlight the net productive systems, which are potentially sustained by intermittent nutrient supply from deep water (Michaels et al., 1994; Dore et al., 2003; Johnson et al., 2010). Deep water contains high amounts of nutrients such as nitrate ($NO_3$), phosphate ($PO_4$), and silicic acid ($Si(OH)_4$), and their supply into the surface ocean alleviates temporarily phytoplankton nutrient stress. Several ship-based experimental studies indicate that deep water additions to subtropical surface waters have induced phytoplankton blooms (Mahaffey et al., 2012; Lampe et al., 2019; Robidart et al., 2019). These studies highlight the shifts in phytoplankton community structure, growth characteristics, and gene expression during the bloom development. However, although nutrient assimilation characteristics are important mechanisms driving the net production, they were not fully described, for example the drawdown ratios (e.g., $\Delta NO_3:\Delta PO_4$ and $\Delta Si(OH)_4:\Delta NO_3$), in these studies.

Phytoplankton N:P stoichiometry is generally based on the canonical ratio of 16 (Redfield, 1958). However, subtropical phytoplankton have higher N:P cellular ratios than Redfield and its mean value for subtropical waters is 28 (Martiny et al., 2013). This higher ratio suggests that subtropical phytoplankton assimilate nutrients with higher N:P ratios than 16. If subtropical phytoplankton assimilate the upwelled deep water nutrients which have nearly Redfield $NO_3:PO_4$ ratios (~16, Fanning, 1992), $PO_4$-excess waters would remain at the surface. The $PO_4$ anomaly ($P^*=PO_4-NO_3/16$) in the upper 120 m of the water column has indeed positive values throughout the subtropical oceans (Deutsch et al., 2007). As with $N^*$ (Gruber and Sarmiento, 1997; Deutsch et al., 2001), $P^*$ is recognized to be controlled by dinitrogen ($N_2$) fixation and denitrification (Deutsch et al., 2007), but it may also be influenced by phytoplankton uptake of the upwelled deep water nutrients.

Subtropical phytoplankton utilize not only $NO_3$ and $PO_4$ but also nitrite ($NO_2$), ammonium ($NH_4$), dissolved organic N (DON), and dissolved organic P (DOP). Amongst them, the concentrations of DON and DOP are one to three orders of magnitude higher than those of dissolved inorganic N (DIN: the sum of $NO_3$, $NO_2$, and $NH_4$) and $PO_4$ in subtropical surface waters (Karl and Björkman, 2015; Sipler and Bronk, 2015). The majority of DON and DOP is likely refractory, but the





bioavailable forms such as urea, amino acid, and ATP play important roles in sustaining primary production in the inorganic
nutrient-depleted subtropical waters (Kanda et al., 1985; Zubkov et al., 2004; Casey et al., 2009; Hill et al., 2011; Shilova et
al., 2017; Björkman et al., 2018). Thus, the dynamics of alternative nutrients other than $NO_3$ and $PO_4$ should be considered
when examining the nutrient assimilation characteristics of subtropical phytoplankton blooms.

Along with N and P assimilation, Si is also assimilated during diatom blooms. Several field studies reported anomalous
$Si(OH)_4$ removal relative to $NO_3$ and $PO_4$ at the sites of diatom blooms in the subtropical oceans (Benitez-Nelson et al.,
2007; Hashihama et al., 2014). These $Si(OH)_4$ removals were not accompanied by stoichiometrically equivalent N and P
removals as with a typical Si:N:P ratio of 16:16:1 (Redfield, 1958; Brzezinski, 1985). Given the linkages between Si and
other elemental cycles, it is important to understand Si dynamics in the subtropical oceans. However, the Si dynamics cannot
be fully explored from snapshot observation in the field (Hashihama et al., 2014), to understand them, experimental
validations are required.

In this study, our aim was to reveal N, P, and Si assimilation characteristics of subtropical phytoplankton blooms as
induced by deep water additions. The onboard bottle incubation experiments were conducted across the subtropical Pacific
Ocean, which has a large geographical variation in surface $PO_4$ from very low (<10 nM) in the western North Pacific (WNP)
to high (>100 nM) in the eastern South Pacific (ESP) (Hashihama et al., 2019; Martiny et al., 2019). Since subtropical
phytoplankton respond to nanomolar increases in nutrient concentrations (Garside, 1985; Eppley and Renger, 1988; Eppley
et al., 1990), we used sensitive liquid waveguide spectrophotometry for measuring nanomolar $NO_3$, $NO_2$, $NH_4$, $PO_4$, and
$Si(OH)_4$. The nanomolar nutrient data enabled us to calculate accurate stoichiometric ratios for N, P, and Si. Along with the
inorganic nutrients, we also examined DON and DOP variations during the incubation experiments. These deep-water
addition experiments successfully induced phytoplankton blooms, while the nutrient drawdown ratios showed geographical
patterns concomitant with the surface $PO_4$ distributions. Here we conclude by discussing the mechanism of the regionally
different drawdown ratios and its possible influences on P* distribution in the subtropical Pacific Ocean.

## 2 Materials and methods

### 2.1 Study areas and water sampling

Observations were conducted at seven stations in the subtropical North and South Pacific Ocean (Table 1 and Fig. 1). Station
A in the WNP was occupied in July 2010 during the R/V Tansei Maru KT-10-13 cruise. Stations 2-21 were occupied along
the transect from the WNP (2 and 5) to the ESP (15, 18, and 21) through the central North Pacific (CNP, 8) for the period
from December 2011 to January 2012 during the R/V Hakuho Maru KH-11-10 cruise. Water sampling was performed using
a conductivity-temperature-depth (CTD) system (Sea-Bird Electronics) equipped with HCl-cleaned Teflon-coated Niskin-X
bottles (General Oceanics). Water samples for incubation experiments were collected from 10 m depth (hereafter referred to
as 'surface') at all stations. At Stations A and 2 in the WNP, deep water from 1500 m depth was also collected.

**Table 1.** Details of the incubation experiments.

| Region | Sampling station | Latitude | Longitude | Sampling date (GMT) | Sampling depth (m) | Incubation period (h) | Mean PAR during incubation (µmol photons m$^{-2}$ s$^{-1}$) |
|---|---|---|---|---|---|---|---|
| WNP | A | 30.00° N | 137.01° E | 2010/7/11 | 10 | 52 | 582 |
| WNP | 2 | 23.00° N | 160.00° E | 2011/12/6 | 10 | 96 | 271 |
| WNP | 5 | 23.00° N | 180.00° E/W | 2011/12/12 | 10 | 96 | 284 |
| CNP | 8 | 22.77° N | 158.09° W | 2011/12/18 | 10 | 48 | 240 |
| ESP | 15 | 23.00° S | 120.00° W | 2012/1/7 | 10 | 96 | 542 |
| ESP | 18 | 30.00° S | 107.00° W | 2012/1/13 | 10 | 96 | 627 |
| ESP | 21 | 23.00° S | 100.00° W | 2012/1/17 | 10 | 96 | 549 |

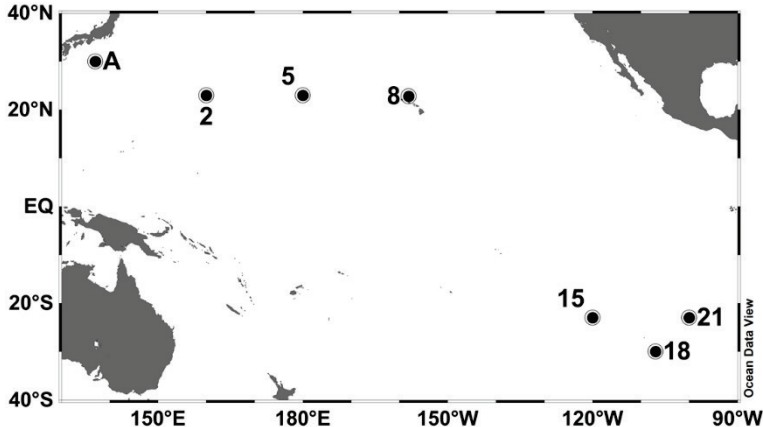

**Figure 1.** Study areas and sampling stations in the subtropical Pacific Ocean. Station A was occupied in July 2010 during the R/V Tansei Maru KT-10-13 cruise. Stations 2-21 were occupied for the period from December 2011 to January 2012 during the R/V Hakuho Maru KH-11-10 cruise.

**2.2 Deep-water addition incubation experiments**

Surface waters collected from 10 m depth were poured into 2.3 L HCl-cleaned polycarbonate bottles and then 50 mL of deep
water was immediately added to the triplicate bottles (2.1% v/v deep water addition). The deep waters collected at Stations A
and 2 were added to the surface water samples at Stations A and 2-21, respectively. The deep waters for the experiments at
Stations 5-21 were kept frozen (-20 °C) until used at each station. Nutrient concentrations in the deep waters at Stations A
and 2 were 37.1 and 39.0 µM DIN, 2.1 and 2.9 µM PO$_4$, and 134.5 and 140.5 µM Si(OH)$_4$, respectively. For all experiments,
triplicate control bottles were prepared. Both the treated and control bottles were incubated for 48-96 h (Table 1) in an on-
deck incubator with flowing surface seawater, which was shaded with appropriate sheeting to give 30% of full sunlight so as
to mimic the ambient photosynthetically active radiation (PAR) condition at 10 m depth. The ambient PAR on deck was
continuously monitored by a LI-COR quantum sensor (LI-190R) with a data logger (LI-1400), and its mean values during



the incubation periods including day and night times are presented in Table 1. During the incubation periods, the bottles were subsampled 5 times for nanomolar nutrients, 3 times for DON and DOP, and 1 time (at the end point) for

phytoplankton. Initial phytoplankton samples at time-zero were collected in single or duplicate directly from the Niskin-X bottles. The DON and DOP samples were not collected at Station A during the KT-10-13 cruise. To assess any significant decrease or increase in the concentrations of nanomolar nutrients, DON, and DOP during the incubation periods, linear regression analyses were performed. In addition, Student $t$-test was performed to determine significant differences between the measured parameter values. In this paper, the significance is reported where $p<0.05$.

**2.3 Determinations of nanomolar nutrients**

Water samples for nanomolar nutrients were collected in 30 mL HCL-cleaned polypropylene tubes and were frozen at -20 °C until analysis. The concentrations of $NO_3$, $NO_2$, $NH_4$, $PO_4$, and $Si(OH)_4$ were determined using an automated liquid waveguide spectrophotometric system equipped with 50-100 cm liquid waveguide capillary cells (LWCC, World Precision Instruments) (Hashihama et al., 2009; Hashihama and Kanda, 2010; Hashihama et al., 2014; Hashihama et al., 2015). The

detection limits for $NO_3$, $NO_2$, $NH_4$, $PO_4$, and $Si(OH)_4$ were 3, 2, 6, 3, and 11 nM, respectively. Although we collected triplicate water samples from the incubated bottles, triplicate analytical data were not available for several samples.

**2.4 Determinations of DON and DOP**

Water samples for DON and DOP were collected in HCL-cleaned polypropylene tubes after removing particulate matter by filtering through pre-combusted Whatman GF/F filters. The samples were frozen at -20 °C until analysis. Total dissolved N

(TDN) and P (TDP) were quantified by a persulfate oxidation method (Hansen and Koroleff, 1999) with a QuAAtro TN-TP analyser (SEAL Analytical) (Yasui et al. 2016; Tamura-Yasui et al., submitted). Concentration of DON was derived from the difference between TDN and DIN concentrations, and that of DOP was derived from the difference between TDP and $PO_4$ concentrations. As with the nutrients (2.3), triplicate analytical data on DON and DOP were not obtained for some samples.

**2.5 Phytoplankton pigment analysis**

Phytoplankton pigment analysis was performed using high-performance liquid chromatography (HPLC). Water volumes of 440-3000 mL were filtered onto GF/F filters, and the filter samples were immediately frozen in liquid nitrogen and stored in a deep freezer (-80 °C) until analysis. Pigment analysis was conducted using the method of Zapata et al. (2000) with a HPLC system (Hashihama et al., 2008). Six phytoplankton pigments - chlorophyll *a* (Chl *a*), divinyl chlorophyll *a* (DVchl *a*), 19´-

butanoyloxyfucoxanthin (But-fuco), fucoxanthin (Fuco), 19´-hexanoyloxyfucoxanthin (Hex-fuco), and zeaxanthin (Zea) - were quantified from the peak area calibrated against that of standard pigments (DHI Water and Environment). Total chlorophyll *a* (Tchl *a*: the sum of Chl *a* and DVchl *a*) was used as an index of total phytoplankton biomass. Total



fucoxanthin (Tfuco: the sum of But-fuco, Fuco, and Hex-fuco) was used as a representative marker of eukaryotic phytoplankton. Zea and DVchl *a* were markers of cyanobacteria and *Prochlorococcus*, respectively.

### 2.6 Microscopic analysis of phytoplankton

Water volumes of 100-1000 mL seawater samples were fixed with neutralized formalin at a final concentration of 1% (v/v). The fixed samples were concentrated through sedimentation in a land-based laboratory. Diatoms were identified and enumerated under an inverted microscope (Utermöhl, 1958).

### 2.7 P* determination using nanomolar nutrient data

Surface P* at the experimental stations was calculated using the measured nanomolar $PO_4$ and DIN data through an equation P*=$PO_4$-DIN/16. In addition, to reveal basin-wide distribution of surface (≤10 m) P* over the oligotrophic Pacific area (40° N-40° S), we assembled nanomolar (<1000 nM) data sets of $PO_4$ and $NO_3$ plus $NO_2$ (N+N), most of which were previously published by the authors in this study (Hashihama et al., 2009; Shiozaki et al., 2009; Hashihama et al., 2010; Sato et al., 2010; Shiozaki et al., 2010; Girault et al., 2013; Sato et al., 2013; Hashihama et al., 2014; Shiozaki et al., 2014; Girault et al., 2015; Sato et al., 2015; Sato et al., 2016; Shiozaki et al., 2016; Shiozaki et al., 2017; Ellwood et al., 2018; Shiozaki et al., 2018; Hashihama et al., 2019; Martiny et al., 2019; Sato and Hashihama, 2019; Yamaguchi et al., 2019; Hashihama et al. submitted; Jiang et al. submitted; Yamaguchi et al. submitted). We also included several unpublished data sets collected by F. Hashihama and T. Kodama. These data sets were obtained by using the liquid waveguide spectrophotometry for $PO_4$ and N+N (Woodward, 2002; Hashihama et al., 2009). Since surface $NH_4$ concentrations were typically low at the sub-nanomolar level and the $NH_4$ data were relatively limited compared to the $PO_4$ and N+N data, we did not use the $NH_4$ data to show the basin-wide distribution of P*. Thus, the P* in this case was calculated through an equation P*=$PO_4$-N+N/16.

### 3 Results

### 3.1 Initial conditions

High temperature (21.76-26.91 °C) and high salinity (34.07-36.50) in the surface waters (10 m) of the seven experimental stations indicated that typical subtropical oceanic waters prevailed in the study regions (Table 2). DIN concentrations at the surface were consistently lower than 50 nM, while $PO_4$ concentrations varied geographically and were extremely low in the WNP (<10 nM; Stations A, 2, and 5), intermediate in the CNP (53 nM; Station 8), and high in the ESP (>100 nM; Stations 15, 18, and 21). Surface P* at these stations showed a geographical variation similar to $PO_4$ concentrations, and this trend was due to the low concentrations of DIN found at those stations. $Si(OH)_4$ concentrations were higher in the WNP and CNP (767-1276 nM; Stations A, 2, 5, and 8) than the ESP (427-541 nM; Stations 15, 18, and 21), and DON and DOP concentrations ranged from 3.47 to 4.45 μM and 0.10 to 0.17 μM, respectively.





Tchl $a$ concentrations at the surface were less than 129 ng L$^{-1}$ with extremely low values at Stations 18 and 21 in the ESP (18 and 3 ng L$^{-1}$, Table 2). Tfuco, Zea, and DVchl $a$ all showed geographical variations similar to Tchl $a$. Tfuco:Zea ratios were lower than 1.0 except for Station 18 in the ESP (2.6), where the biomass proportion of eukaryotes to
175   cyanobacteria was relatively high. DVchl $a$:Tchl $a$ ratios (indices of the contribution of *Prochlorococcus* to total phytoplankton) were mostly 0.4-0.5, but the lower ratios were observed at Stations 18 and 21 in the ESP (0.1 and 0.2). Cell densities of surface diatoms were consistently low (3-88 cells L$^{-1}$) at all stations.



**Table 2.** Initial conditions for incubation samples collected from 10 m depth. nd: no data.

| Region | Station | Temperature (°C) | Salinity | DIN (nM) | PO$_4$ (nM) | P* (nM) | Si(OH)$_4$ (nM) | DON (µM) | DOP (µM) | Tchl $a$ (ng L$^{-1}$) | Tfuco (ng L$^{-1}$) | Zea (ng L$^{-1}$) | DVchl $a$ (ng L$^{-1}$) | Tfuco :Zea (g:g) | DVchl $a$ :Tchl $a$ (g:g) | Diatoms (cells L$^{-1}$) |
|---|---|---|---|---|---|---|---|---|---|---|---|---|---|---|---|---|
| WNP | A | 26.91 | 34.07 | 9 | 7 | 6 | 1276 | nd | nd | 45 | 17 | 32 | 18 | 0.5 | 0.4 | 88 |
| WNP | 2 | 26.72 | 34.92 | 16 | 2 | 1 | 870 | 3.47 | 0.17 | 26 | 13 | 27 | 13 | 0.5 | 0.5 | 20 |
| WNP | 5 | 26.37 | 35.31 | 11 | 8 | 7 | 767 | 3.65 | 0.19 | 52 | 21 | 32 | 24 | 0.6 | 0.5 | 18 |
| CNP | 8 | 24.24 | 35.29 | 22 | 53 | 52 | 989 | 4.25 | 0.21 | 129 | 44 | 47 | 65 | 0.9 | 0.5 | 4 |
| ESP | 15 | 25.28 | 36.50 | 16 | 228 | 227 | 541 | 4.45 | 0.10 | 49 | 17 | 46 | 25 | 0.4 | 0.5 | 20 |
| ESP | 18 | 21.76 | 35.57 | 47 | 124 | 121 | 427 | 3.51 | 0.15 | 18 | 11 | 4 | 3 | 2.6 | 0.1 | 3 |
| ESP | 21 | 24.09 | 35.87 | 40 | 272 | 270 | 439 | 3.62 | nd | 3 | 2 | 5 | 1 | 0.4 | 0.2 | 19 |





## 3.2 Phytoplankton response to deep water additions

Mean Tchl $a$ concentrations were three to ten times higher in the treated bottles than observed for the control bottles in all experiments, although no significant difference was observed at Station A due to a highly variable results in the treated bottles (Fig. 2a). These trends in increasing Tchl $a$ indicate that the deep water additions positively induced phytoplankton blooms. The Tchl $a$ differences between the treated and control bottles were greatest at Station A in the WNP (178 ng L$^{-1}$) and Station 15 in the ESP (306 ng L$^{-1}$), and net growth rates of the blooms (as Tchl $a$, relative to the control) were higher at these two stations (0.70 and 0.58 d$^{-1}$) than other stations (0.27-0.50 d$^{-1}$). Mean Tfuco concentrations were significantly higher in the treated than in the control bottles except for Station A (Fig. 2b). Mean Zea concentrations were significantly higher in the treated bottles in the WNP (Stations A, 2, and 5) and at two sites of the ESP (Stations 15 and 21) (Fig. 2c). Mean DVchl $a$ concentrations were significantly higher in the treated bottles at two sites in the WNP (Stations 2 and 5) and Station 21 in the ESP (Fig. 2d). The pigment concentrations in the control bottles were similar to those in the initial conditions, indicating that changes in pigment concentrations due to photoacclimation during the incubation periods are small.

Mean Tfuco:Zea ratios were higher in the treated than in the control bottles, although no significant differences were observed at Stations A and 18 (Fig. 2e). The higher Tfuco:Zea ratios in the treated bottles imply that biomass increases in eukaryotes were relatively large compared to those of cyanobacteria. The ratios of Tfuco:Zea in the treated bottles were higher in the ESP (Stations 15, 18, and 21) than the WNP and CNP (Stations A, 2, 5, and 8), indicating that the proportions of eukaryotes (cyanobacteria) were higher (lower) in the ESP than the WNP and CNP. Mean DVchl $a$:Tchl $a$ ratios were significantly lower in the treated bottles than the control, except for Stations 18 and 21 in the ESP, where the ratios in the control were quite low (<0.1) as observed in the initial conditions (Fig. 2f). The dominance of *Prochlorococcus* in the subtropical waters was no longer evident following the deep water additions.

Cell densities of diatoms were significantly higher in the treated bottles at two sites of the WNP (Stations A and 2) and Station 18 in the ESP (Fig. 2g). An exceptionally high mean density of diatoms (907 cells L$^{-1}$), mostly consisting of *Nitzschia longissima*, was observed in the treated bottles at Station 15 in the ESP, although no significant difference between the densities in the control and treated bottles was seen.



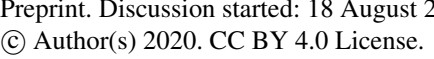





**Figure 2.** (a) Tchl *a*, (b) Tfuco, (c) Zea, (d) DVchl *a*, (e) Tfuco:Zea ratios, (f) DVchl *a*:Tchl *a* ratios, and (g) diatom cell densities in the control and treated bottles after 48-96 h incubations at seven stations. A grey horizontal line in each panel indicates a mean or single concentration of each pigment at the initial point. Error bars denote standard deviations. Significant differences (*t*-test, $p<0.05$) between the values in the control and treated bottles are depicted by asterisks. Net growth rates, μ ($d^{-1}$) of the phytoplankton blooms (as Tchl *a*, relative to the control) are denoted in (a) Tchl *a*. In (g) diatom cell densities, a scale of y-axis at Station 15 (~1500 cells $L^{-1}$) is different from those at other stations (~300 cells $L^{-1}$).

### 3.3 Nutrient drawdown

Following the phytoplankton blooms, DIN and $PO_4$ concentrations in the treated bottles at all stations showed significant linear decreases as a function of the incubation times ($r^2>0.82$, Figs. 3a and 3b). In contrast, the concentrations in the control bottles at all stations showed no significant trends. The DIN decreases were largely ascribed to $NO_3$ decreases (Fig. A1a), and interestingly, $NO_2$ concentrations in the treated bottles at all stations significantly increased with time (Fig. A1b). There were also significant linear increases in the control bottles for $NO_3$ at Stations 8 and 15, and $NO_2$ at Station 18 but these changes were quite small (<13 nM). $NH_4$ concentrations in the treated and control bottles showed no significant trends with time, but occasionally high standard deviations (>100 nM) were observed (Fig. A1c).

Net assimilation rates (slopes of linear regression lines in Figs. 3a and 3b) of DIN and $PO_4$ in the treated bottles varied from 1.38 to 7.34 nmol N $L^{-1}$ $h^{-1}$ and from 0.133 to 0.713 nmol P $L^{-1}$ $h^{-1}$, respectively. These rates were relatively low (<1.49 nmol N $L^{-1}$ $h^{-1}$ and <0.154 nmol P $L^{-1}$ $h^{-1}$, respectively) in the WNP during winter (December, Stations 2 and 5) where relatively low mean PAR was observed during the incubation periods (≤284 μmol photons $m^{-2}$ $s^{-1}$, Table 1). Differences between the control-corrected mean concentrations of DIN and $PO_4$ at the start and end points of the incubation (ΔDIN and $ΔPO_4$) varied from 123 to 750 nM and from 9 to 53 nM, respectively (Table 3). The values of ΔDIN and $ΔPO_4$ normalized by the incubation times (h) were almost identical to the net assimilation rates of DIN and $PO_4$ in the treated bottles ($r^2=0.99$ and $r^2=0.91$, respectively).

Unlike the DIN and $PO_4$ concentrations, $Si(OH)_4$ concentrations in the treated bottles at all stations did not show any significant linear decreases with the occasionally high standard deviations (>500 nM) (Fig. 3c). The insignificant trends in $Si(OH)_4$ concentrations were also observed in the control bottles at all stations. Difference between the control-corrected mean $Si(OH)_4$ concentrations at the start and end points ($ΔSi(OH)_4$) showed net drawdown values ranging from 7 to 464 nM, although mean $Si(OH)_4$ concentrations in the treated bottles at the start and end points were not significantly different except at Station A (Table 3).



**Figure 3.** Temporal changes in concentrations of (a) DIN, (b) PO$_4$, and (c) Si(OH)$_4$ in the control (blue) and treated (red) bottles during the incubation periods at seven stations. Error bars denote standard deviations. Duplicate or single data are denoted as mean or single values without error bars. Linear regression lines are depicted when significant decreases ($p<0.05$) in the mean concentrations against time were observed.





**Table 3.** Nutrient drawdowns and their ratios throughout the incubation periods. [a] Difference between the control-corrected
mean concentrations at start and end points of incubations. [b] No significant difference between the mean $Si(OH)_4$
concentrations in the treated bottles at start and end points of incubations (*t*-test, $p > 0.05$).

| Region | Station | Incubation period (h) | $\Delta DIN$[a] (nM) | $\Delta PO_4$[a] (nM) | $\Delta Si(OH)_4$[a] (nM) | $\Delta DIN:\Delta PO_4$ (mol:mol) | $\Delta Si(OH)_4:\Delta DIN$ (mol:mol) |
|---|---|---|---|---|---|---|---|
| WNP | A | 52 | 280 | 22 | 100 | 12.9 | 0.36 |
| WNP | 2 | 96 | 123 | 15 | 21[b] | 8.0 | 0.17 |
| WNP | 5 | 96 | 156 | 15 | 464[b] | 10.7 | 2.97 |
| CNP | 8 | 48 | 158 | 11 | 304[b] | 14.4 | 1.93 |
| ESP | 15 | 96 | 750 | 53 | 374[b] | 14.1 | 0.50 |
| ESP | 18 | 96 | 277 | 9 | 7[b] | 29.6 | 0.03 |
| ESP | 21 | 96 | 232 | 14 | 41[b] | 16.2 | 0.18 |

### 3.4 Nutrient drawdown ratio

In all experiments, the enriched nutrients in the treated bottles were not fully taken up by phytoplankton during the
incubation periods (Fig. 3). Therefore, we assessed the nutrient drawdown ratios using $\Delta DIN$, $\Delta PO_4$, and $\Delta Si(OH)_4$ (Table 3).
$\Delta DIN:\Delta PO_4$ ratios varied from 8.0 to 29.6, and relatively lower ratios ($\leq 12.9$) were observed in the WNP (Stations A, 2, and
5). $\Delta Si(OH)_4:\Delta DIN$ ratios varied from 0.03 to 2.97 with most stations less than 1 except for Stations 5 and 8. However, these
ratios, except at Station A (0.36), involved uncertainties due to no significant decreases in $Si(OH)_4$ concentrations in the
treated bottles during the incubations.

We also evaluated the drawdown characteristics using the control-corrected mean concentrations of DIN and $PO_4$ at
sampling points during the incubation periods. A plot of $PO_4$ against DIN showed strong negative linear relationships
($r^2 > 0.76$), except for Station 21 ($r^2 = 0.23$) (Fig. 4). Here, the drawdown ratio of DIN to $PO_4$ was expressed as 1/slope of the
linear regression, and ranged from 10.2 to 31.7. These drawdown ratios were almost identical to the $\Delta DIN:\Delta PO_4$ ratios in
Table 3 ($r^2 = 0.97$). In addition, we observed unique variations in $PO_4$-intercepts of the linear regression lines. The $PO_4$-
intercepts varied from -23 to 34 nM, and the relatively lower values ($< -10$ nM) being observed in the WNP (Stations A, 2,
and 5).





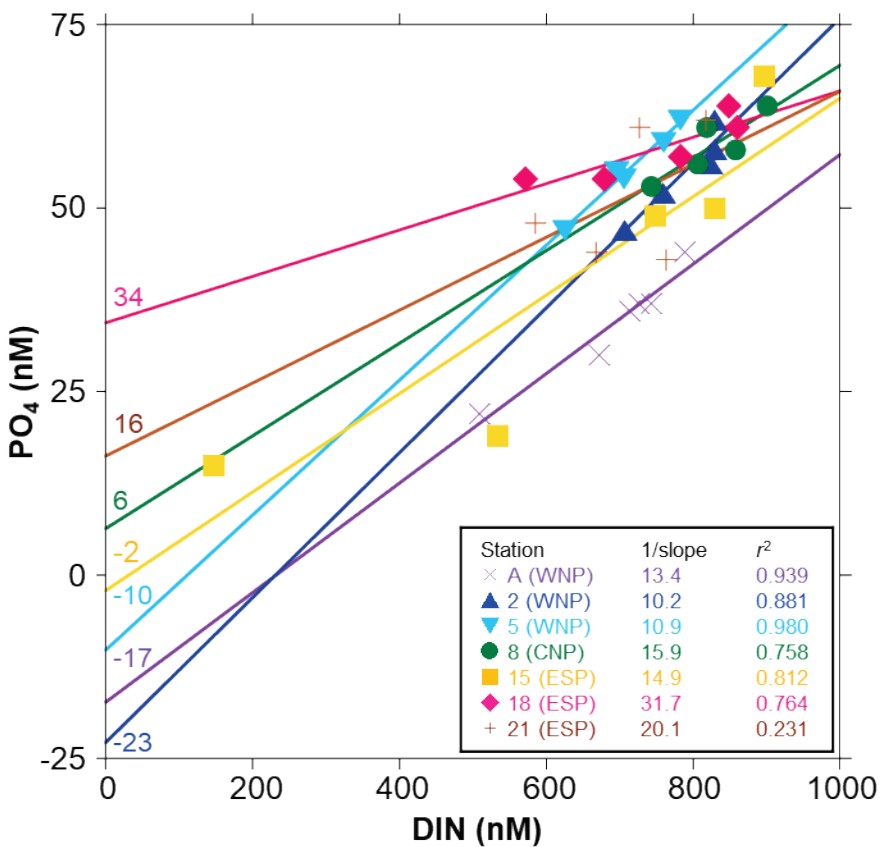

**Figure 4.** Scatter plots of the control-corrected mean concentrations of PO$_4$ against DIN in the incubation experiments at seven stations. Linear regression lines with their parameters (1/slope, PO$_4$-intercept, and $r^2$) at seven stations were denoted by the different colours.

## 3.5 DON and DOP

DON and DOP concentrations in both the treated and control bottles did not show any significant increase or decrease as a function of the incubation time, except for DOP in the treated bottles at Station 8 in the CNP (Figs. A2a and A2b). At this station, the DOP concentrations in the treated bottles significantly increased with time, although the change over 48 h (0.03 μM) was smaller than that from 0 to 24 h (-0.07 μM) in the control bottles. Overall, the DON and DOP concentrations in the treated bottles were similar to those in the control bottles, indicating that the deep water additions did not alter DON and DOP regimes during the incubation periods.



### 3.6 Basin-wide P* distribution in the oligotrophic Pacific

The assembled surface N+N and $PO_4$ data (Figs. A3a and A3b) revealed a detailed surface P* distribution over the

oligotrophic Pacific Ocean (Fig. 5). The distributional pattern of P* was similar to that of $PO_4$ (Fig. A3b), mainly due to a

low N+N area (<100 nM) (Fig. A3a). The P* showed a clear west-east gradient from <50 nM in the western basin to ~500

nM in the eastern basin. In the western basin, the extremely low P* (<10 nM) was found in the WNP. Stations A, 2, and 5

were located within this low P* area. Station 8 was in the intermediate P* area (50-100 nM) in the CNP, while Stations 15,

18, and 21 were within the high P* area (>100 nM) in the ESP.


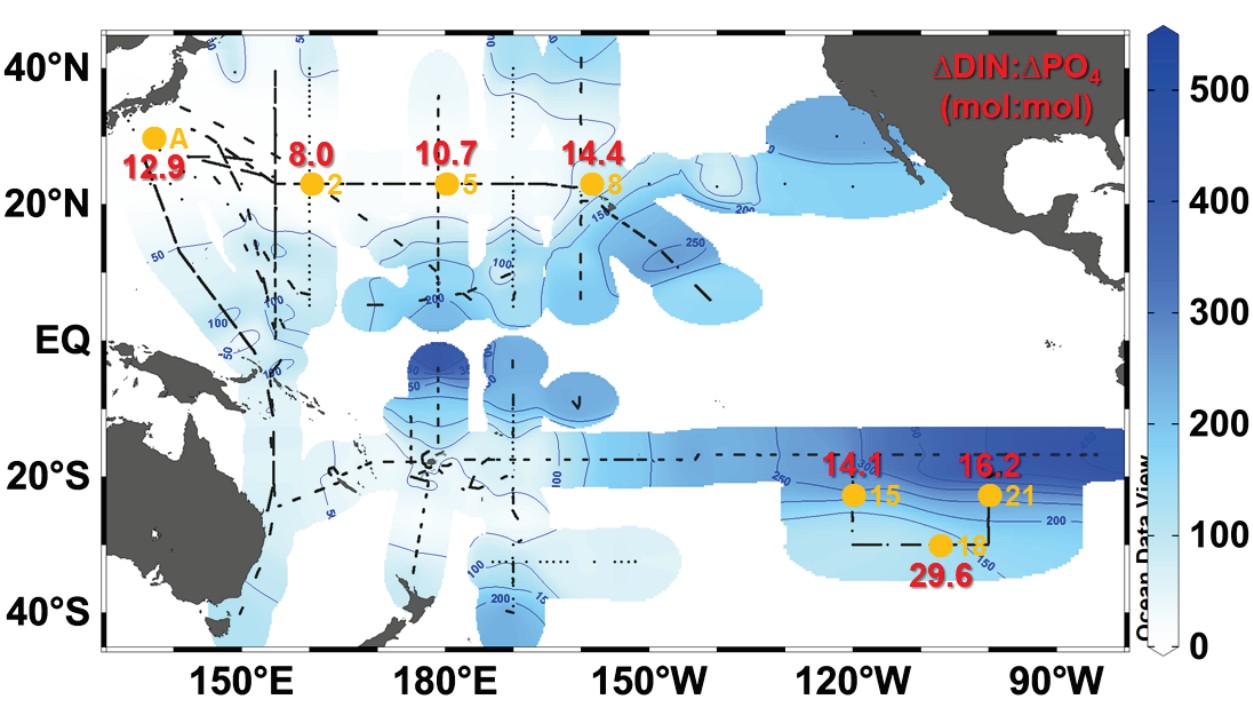

**Figure 5.** Surface distributions of P* in the oligotrophic Pacific Ocean (40° N-40° S). The P* data were derived from the

assembled data on nanomolar concentrations (<1000 nM) of $PO_4$ and N+N. Small black dots denote sampling stations for the

nanomolar $PO_4$ and N+N. Large orange circles denote the stations where the incubation experiments were conducted. Red

values indicate the experimentally-determined ΔDIN:ΔPO₄ ratios (Table 3).





## 4 Discussion

### 4.1 Phytoplankton blooms following deep water additions

Our study confirms that deep water additions induced phytoplankton blooms in various regions of the subtropical Pacific Ocean. Such induced blooms have also been reported at Station ALOHA in the subtropical North Pacific Ocean (Mahaffey

et al., 2012). The ALOHA experiments revealed that Tchl *a*-based growth rates following deep water additions were higher in boreal summer than boreal winter. A similar seasonal trend was observed in the North Pacific stations reported here; net growth rates were higher at the westernmost site in July (Station A, 0.70 d$^{-1}$) than other sites in December (Stations 2, 5, and 8, 0.27-0.50 d$^{-1}$) (Fig. 2a). The low growth rates in winter could be explained by low assimilation rates of nutrients (Fig. 3). These results suggest that the magnitude of a subtropical phytoplankton bloom is regulated by additional seasonal factors

such as PAR (Table 1).

The net phytoplankton growth rates were relatively low at Stations 18 and 21 in the ESP (0.43 and 0.40 d$^{-1}$, respectively, Fig. 2a), even for austral summer conditions with high PAR (Table 1). Since the ESP is known as a low dust deposition area, growth by the resident phytoplankton is considered to be limited by iron (Fe) (Jickells et al., 2005; Blain et al., 2008; Wagener et al., 2008; Moore et al., 2013). However, a Fe-enrichment incubation experiment at Station 18 demonstrated that

there was no significant difference between the Fe-enriched and control bottles for phytoplankton, nutrients, DON, and DOP during a 96 h incubation, and also these parameter values were little changed from their initial values as the initial values lay within standard deviations of the mean values in the Fe-enriched and control bottles (Appendix methods, Table A1). Similar experimental results were reported by Bonnet et al. (2008) and they suggested that phytoplankton growth in the ESP is limited by N rather than Fe. Furthermore, the deep waters used in this study were from a depth (1500 m) within North

Pacific intermediate water that typically contains high dissolved Fe (>0.6 nM, i.e., not iron-limiting) (Nishioka et al., 2013; Nishioka et al., 2020). These lines of evidence imply that surface phytoplankton in the ESP and their bloom formation were not primarily limited by Fe. Although grazing by zooplankton is a possible factor controlling phytoplankton net growth, it was suggested to be not strong in the case of the subtropical phytoplankton blooms following deep water additions (Mahaffey et al., 2012). There is further research required to understand the factors controlling the bloom development in the

ESP.

The induced phytoplankton blooms in this study were accompanied by changes in community structure. Several nutrient enrichment experiments conducted in the subtropical oceans have demonstrated significant increases in eukaryotic phytoplankton following nutrient enrichments, particularly of N (Bonnet et al., 2008; Moore et al., 2008; Mahaffey et al., 2012; Shilova et al., 2017; Rii et al., 2018; Lampe et al., 2019; Robidart et al., 2019). Similar blooms dominated by

eukaryotic phytoplankton were also observed at most stations in this study as evidenced by the Tfuco increases (Fig. 2b). In addition, significant increases in cyanobacteria (Zea and DVchl *a*) following deep water additions were also observed particularly in the WNP and CNP (Figs. 2c and d). The relative proportions of cyanobacteria to eukaryotes in the treated bottles were lower in the ESP than the WNP and CNP (Fig. 2e). At Stations 18 and 21 in the ESP, the low proportions of



*Prochlorococcus* at time zero (Table 2) might influence the low proportions of cyanobacteria in the induced phytoplankton

blooms. On the other hand, the relative increases of cyanobacterial abundances in the North Pacific experiments are likely driven by seasonal phytoplankton response to nutrient enrichment. Mahaffey et al. (2012) reported that there were less increases in eukaryote abundances in winter than in summer at Station ALOHA following deep water additions. Because of this, the relative increases in cyanobacteria in the winter-time North Pacific (Stations 2, 5, and 8) might be significant. However, although Station A in the WNP was occupied in summer, we did not observe any significant eukaryotic bloom

following deep water addition (Fig. 2e). This opposing trend at Station A could be due to regional differences of seasonal phytoplankton responses to nutrient enrichments, as summer eukaryotic blooms frequently occur in the eastern basin compared to the western basin in the North Pacific (Wilson, 2011; Villareal et al., 2012; Hashihama et al., 2014; Jiang et al. submitted).

**4.2 N and P drawdown characteristics**

Our incubation experiments have revealed consistent linear decreases in DIN and $PO_4$ concentrations, at nanomolar levels, along with the development of phytoplankton blooms (Figs. 3a and 3b). Additionally, accurate measurements of nanomolar inorganic N species detected the consistent increases in $NO_2$ concentrations in the treated bottles in which there were large decreases in $NO_3$ concentrations. This trend was particularly prominent at Station A (Figs. A1a and A1b). These $NO_2$ increases could be due to in vitro nitrification of $NH_4$ and/or incomplete assimilation of $NO_3$ by phytoplankton (Lomas and

Lipschultz, 2006). While the factors controlling regional or seasonal differences in these $NO_2$ increases remains unknown, we demonstrate that sensitive measurements for multiple nutrients enable us to detect trace, but important, biogeochemical dynamics in response to the addition of nutrient-rich deep water.

Unlike the DIN and $PO_4$, DON and DOP concentrations did not show any consistent changes over time despite the occurrence of the phytoplankton blooms (Fig. A2). Although natural phytoplankton in the nutrient-depleted oligotrophic

oceans show high affinity to DON and DOP (Karl and Björkman, 2015; Sipler and Bronk, 2015), the DON and DOP concentrations here did not indicate net consumption in either the treated or control bottles. For the treated bottles, phytoplankton N and P demands were largely met by the enriched inorganic nutrients - which were not exhausted during the incubation periods (Figs. 3a and 3b). The DON and DOP dynamics during the development of phytoplankton blooms appear to be in an equilibrium between consumption and production, as seen under ambient conditions like those in the control

bottles (Fig. A2).

Although the induced phytoplankton blooms in this study were solely dependent on DIN and $PO_4$, regionally unique $\Delta DIN:\Delta PO_4$ ratios were unveiled (Table 3). The $\Delta DIN:\Delta PO_4$ ratios showed a geographical trend with lower ratios in the $PO_4$-depleted WNP (8.0-12.9) than in the other $PO_4$-replete regions (14.1-29.6). While the ratios in the $PO_4$-replete regions were similar to the range (16-28) from the Redfield ratio to the subtropical particulate N:P ratios (Redfield, 1958; Martiny et

al., 2013), those in the WNP were distinctly lower than 16. Cellular N:P ratios in the subtropical phytoplankton are higher in cyanobacteria (25-35) than for eukaryotes (16) (Martiny et al., 2013). Because the phytoplankton blooms in the WNP were





composed of communities with a relatively high proportion of cyanobacteria (Figs. 2e and 2f), the lower $\Delta DIN:\Delta PO_4$ ratios (8.0-12.9) could not be explained by phytoplankton cellular N:P ratios. Since macro-scale (>2000 km) exhaustion of $PO_4$ in the WNP is coupled with $N_2$ fixation (Hashihama et al., 2009; Martiny et al., 2019), $N_2$ fixation by diazotrophic

cyanobacteria might potentially meet some of phytoplankton N demand. However, assuming a relatively high $N_2$ fixation rate in the WNP (5 nmol N L$^{-1}$ d$^{-1}$, Hashihama et al., submitted), the contributions of $N_2$ fixation to $\Delta(DIN+N_2)$ were small (4-16%), leading to still lower $\Delta(DIN+N_2):\Delta PO_4$ ratios (9.3-13.4) than 16.

Other explanations for the regionally unique $\Delta DIN:\Delta PO_4$ ratios we observed (Table 3) are based on resident communities being 'primed' for rapid P acquisition. Lomas et al. (2014) reported that phytoplankton in the western

subtropical North Atlantic have active $PO_4$ transporters which are rapidly induced under severely $PO_4$-depleted condition. In addition, several studies have reported that microbial genes for high-affinity $PO_4$ transporter (*pstSCAB*) are enriched in the $PO_4$-depleted regions in the western North Atlantic and WNP compared to the $PO_4$-replete regions in the CNP and ESP (Coleman and Chisholm, 2010; Hashihama et al., 2019). These studies indicated that the phytoplankton in the $PO_4$-depleted regions possess a high $PO_4$ uptake capability. Since N perturbation to phytoplankton little alters their cellular N quota

(Moreno and Martiny, 2018), a high cellular P accumulation by the high uptake capability could induce the surplus drawdowns of $PO_4$ relative to DIN in the incubation experiments of the $PO_4$-depleted WNP. Since the DOP concentrations in the treated bottles in the WNP did not increase significantly (Fig. A2), the assimilated $PO_4$ would be sustained in the cellular P components such as polyphosphate, that typically accumulates in particulate P in the $PO_4$-depleted regions such as the western North Atlantic and WNP (Martin et al., 2014; Hashihama et al., submitted).

**4.3 Si and N drawdown characteristics**

Previous field studies have reported that natural diatom blooms in the subtropical North Pacific were accompanied by high $Si(OH)_4:DIN$ drawdown ratios (>1) (Benitez-Nelson et al., 2007; Hashihama et al., 2014). However, in our incubation experiments, most of the $\Delta Si(OH)_4:\Delta DIN$ ratios were less than 1 and involved no significant values of $\Delta Si(OH)_4$ (Table 3). Furthermore, there was no significant $\Delta Si(OH)_4$ even with a large increase (806 cells L$^{-1}$ relative to the control) in diatom

stocks at Station 15 (Table 3 and Figs. 2g and 3c). This increased diatom density is comparable to that in the natural bloom in the WNP reported by Hashihama et al. (2014). This mismatch between increased stocks and little change in $\Delta Si(OH)_4$ implies that $\Delta Si(OH)_4:\Delta DIN$ ratio is not so high in an early stage of diatom blooms. Low $\Delta Si(OH)_4:\Delta DIN$ ratio in the early bloom phase was also observed in a mesoscale Fe enrichment experiment in the northeastern part of the subarctic Pacific, suggesting no Fe limitation of diatoms in the early bloom phase (Boyd et al., 2005). Probably, in a Fe- or DIN-depleted late

stage of the bloom, selective $Si(OH)_4$ removal by diatoms (>1 of $\Delta Si(OH)_4:\Delta DIN$) occurs through putative biogeochemical processes such as selective Si export (Si pump), anomalous Si uptake associated diatom physiology, and/or Si uptake supported by $N_2$ fixation (Dugdale and Wilkerson, 1998; Takeda, 1998; Boyd et al., 2005; Benitez-Nelson et al., 2007; Brzezinski et al., 2011; Krause et al., 2013; Hashihama et al., 2014).





### 4.4 Possible influences of the subtropical phytoplankton blooms on P* distribution

The present study is the first to reveal basin-wide distributions of surface P* in the oligotrophic Pacific Ocean using nanomolar N+N and $PO_4$ data (Fig. 5). The distributional pattern of surface P* coincided with that obtained in the upper 120 m using micromolar $NO_3$ and $PO_4$ data (Deutsch et al., 2007), indicating that a relatively homogenous P* condition prevails throughout the water column from the surface to 120 m depth. Based on the conventional concept, the upper layer P* is likely controlled by $N_2$ fixation and denitrification (Deutsch et al., 2007). However, by comparing surface P* distribution

with experimentally determined ΔDIN:Δ$PO_4$ ratios across the subtropical Pacific, we see insights into additional controls on P*. The low (≤12.9) and high (≥14.1) ΔDIN:Δ$PO_4$ ratios geographically corresponded to the low and high P* in the WNP (<50 nM, Stations A, 2, and 5) and the CNP and ESP (>50 nM, Stations 8, 15, 18, and 21), respectively (Fig. 5).

A comparison of the $PO_4$-intercepts that were determined from the deep-water addition experiments with surface P* (from ambient concentrations) was conducted (Fig. 6). This cross-comparison was valuable since both represent metrics for

excess $PO_4$ that remains after DIN exhaustion. The $PO_4$-intercepts and P* are hereafter referred to as 'bloom P*' and 'ambient P*', respectively. The bloom and ambient P* showed a similar geographical trend both being relatively low in the $PO_4$-depleted WNP (Stations A, 2, and 5) when compared with the $PO_4$-replete regions (Stations 8, 15, 18, and 21). The one to two orders of magnitude higher ambient P* than bloom P* in the $PO_4$-replete regions suggests that, rather than phytoplankton uptake, denitrification (and also anammox) has a more pronounced influence on setting excess $PO_4$ in those

regions. This trend may be particularly important in the ESP which is the vicinity of an oxygen minimum zone with active denitrification and anammox conditions (Paulmier and Ruiz-Pino, 2009). Alternately, $N_2$ fixation may exert an influence on ambient P* in the $PO_4$-depleted WNP. However, several studies reported that directly measured $N_2$ fixation rates are not consistently high in the WNP compared to other subtropical Pacific regions (Shiozaki et al., 2009; Shiozaki et al., 2010; Hashihama et al., submitted). Given that natural phytoplankton blooms in the subtropical oceans have a large impact on

nutrient dynamics through new production (Benitez-Nelson et al., 2007; McGillicuddy et al., 2007; Dore et al., 2008), the surplus $PO_4$ removal by phytoplankton bloom as observed in our experiments might play a significant role in maintaining low ambient P* in the WNP.

Furthermore, we found the unique result that bloom P* showed negative values in the $PO_4$-depleted WNP (Stations A, 2, and 5), while the ambient P* did not exhibit negative values (Fig. 6). These negative bloom P* imply the presence of

alternative P sources other than $PO_4$ to fully exhaust DIN. Since lower DOP concentrations and higher alkaline phosphatase activity were observed in the WNP than other subtropical Pacific regions (Hashihama et al., 2019), active DOP utilization in the WNP likely contributes to the DIN exhaustion. The DOP utilization in the WNP is enhanced at ~10 nM of $PO_4$ concentrations (Sato et al., 2013). Slightly positive ambient P* in the WNP (~10 nM; nearly equal to $PO_4$ concentration due to low DIN concentration, see Table 2) may reflect switching to DOP acquisition mode of phytoplankton P utilization. These

perspectives suggest that, in the studies on subtropical nutrient biogeochemistry using tracers such as N* and P*, the bioavailable fraction of DOP could be an important factor as well as DIN and $PO_4$.



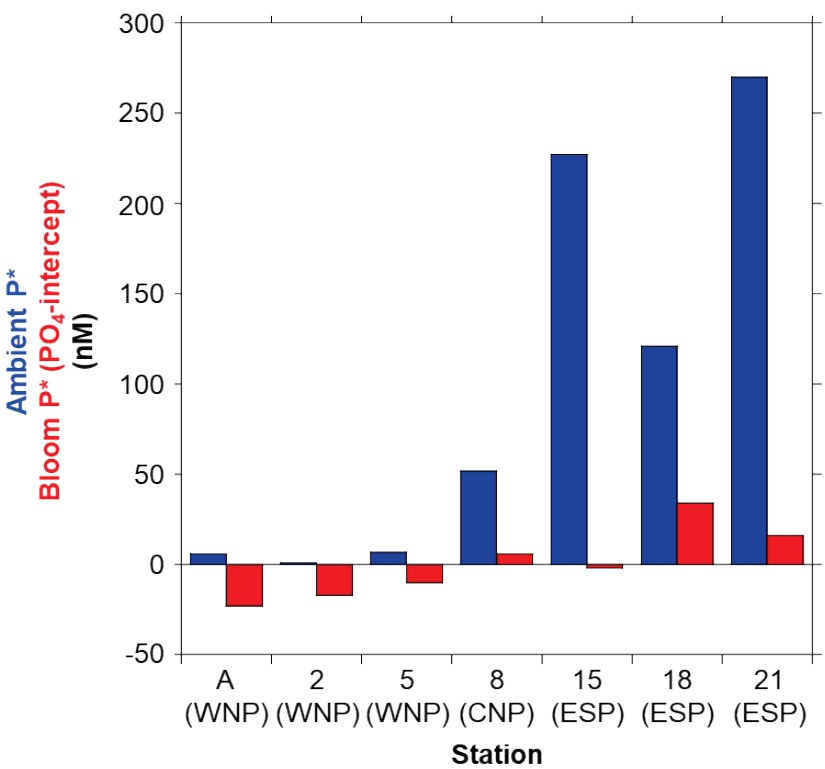

**Figure 6.** Comparison of ambient P* (blue) and bloom P* (red) at seven stations. The bloom P* is identical to the experimentally determined PO$_4$-intercept in Fig. 4.

## 5 Conclusions

By applying highly sensitive analytical methodology, we have revealed nutrient drawdowns and their ratios during the developments of phytoplankton blooms as induced by deep water additions to the surface water of the oligotrophic Pacific Ocean. The ΔDIN:ΔPO$_4$ ratios showed a clear geographical variation from low in the PO$_4$-depleted WNP to high in the PO$_4$-replete ESP. While the ΔDIN:ΔPO$_4$ ratios in the PO$_4$-replete regions were similar to the range from the Redfield ratio to typical subtropical particulate N:P ratio (16-28), those in the PO$_4$-depleted regions (8.0-12.9) could not be expected from the conventional phytoplankton N:P ratios. The lower ΔDIN:ΔPO$_4$ ratios were likely due to the high PO$_4$ uptake capability of low PO$_4$-adapted subtropical phytoplankton. The regional trend in ΔDIN:ΔPO$_4$ ratios was aligned with that of ambient P* in the oligotrophic Pacific. Although it remains necessary to examine nutrient assimilation characteristics in natural phytoplankton blooms, the regional variation in ΔDIN:ΔPO$_4$ ratios as observed in our experiments appears to at least control basin-scale ambient P* distribution in addition to conventional N$_2$ fixation and denitrification (also anammox). We have also demonstrated that accurate measurements of nanomolar nutrients are powerful tools in investigating trace nutrient dynamics.





Further application of these tools to the field and experimental studies would be beneficial for understanding of nutrient biogeochemistry in the oligotrophic ocean.

**Appendices**

**Appendix methods**

The Fe-enrichment incubation experiment was conducted using the surface water (10 m depth) collected at Station 18 in the ESP (Table 1 and Fig. 1). Water sampling was performed using HCl-cleaned Teflon-coated Niskin-X bottles (General Oceanics) on a CTD system (Sea-Bird Electronics) attached to a titanium-armored cable. This sampling procedure succeeded

in avoiding Fe contamination as reported previously (Shiozaki et al., 2018). The surface water was poured into 1.19 L polycarbonate bottles and then Fe was enriched as iron chloride ($FeCl_3$, Iron Standard Solution Fe 1000, Wako) at the final concentration of 1.8 nM. The triplicate bottles for either Fe-enrichment or control were prepared. These bottles were pre-cleaned sequentially with neutral detergent, 1 M HCl, and 0.3 M hot HCl (for Analysis of Poisonous Metals, Wako), and filled with pure water for a day (Takeda and Obata, 1995). Both the Fe-enriched and control bottles were incubated for 96 h

in the on-deck incubator as described in 2.2. After 96 h, the incubated bottles were sampled for nanomolar nutrients, DON, DOP, and Tchl $a$. Initial samples for nanomolar nutrients, DON, DOP, and Tchl $a$ were collected in duplicate directly from the Niskin-X bottles. The samples for nanomolar nutrients, DON, and DOP were processed and analysed as described in 2.3 and 2.4. For Tchl $a$ here, a water volume of 100 mL was filtered onto GF/F filters, and the filter samples extracted with $N,N$-dimethylformamide (DMF, Wako) were analysed using a Turner Design fluorometer (Suzuki and Ishimaru, 1990). Student $t$-

test was performed to determine significant differences ($p<0.05$) between the measured parameter values in the Fe-enriched and control bottles.

**Table A1.** Results of a Fe-enrichment incubation experiment at Station 18 in the ESP. Differences between mean values in control and Fe-treatment were insignificant for all parameters ($t$-test, $p>0.05$).

| Parameter | Initial (0 h) (mean in duplicate) | Control after 96 h (mean±SD, $n$=3) | Fe-treatment after 96 h (mean±SD, $n$=3) |
|---|---|---|---|
| Tchl $a$ (ng L$^{-1}$) | 31 | 29±2 | 30±3 |
| DIN (nM) | 7 | 13±6 | 9±3 |
| PO$_4$ (nM) | 133 | 131±6 | 133±3 |
| Si(OH)$_4$ (nM) | 444 | 438±37 | 467±26 |
| DON (μM) | 3.52 | 3.57±0.12 | 3.63±0.14 |
| DOP (μM) | 0.12 | 0.09±0.03 | 0.09±0.07 |






**Figure A1.** Temporal changes in concentrations of (a) $NO_3$, (b) $NO_2$, and (c) $NH_4$ in the control (blue) and treated (red) bottles during the incubation periods at seven stations. Error bars denote standard deviations. Duplicate or single data are denoted as mean or single values without error bars. Linear regression lines are depicted when significant decreases or increases ($p<0.05$) in the mean concentrations against time were observed.

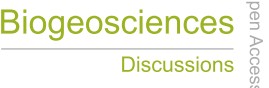

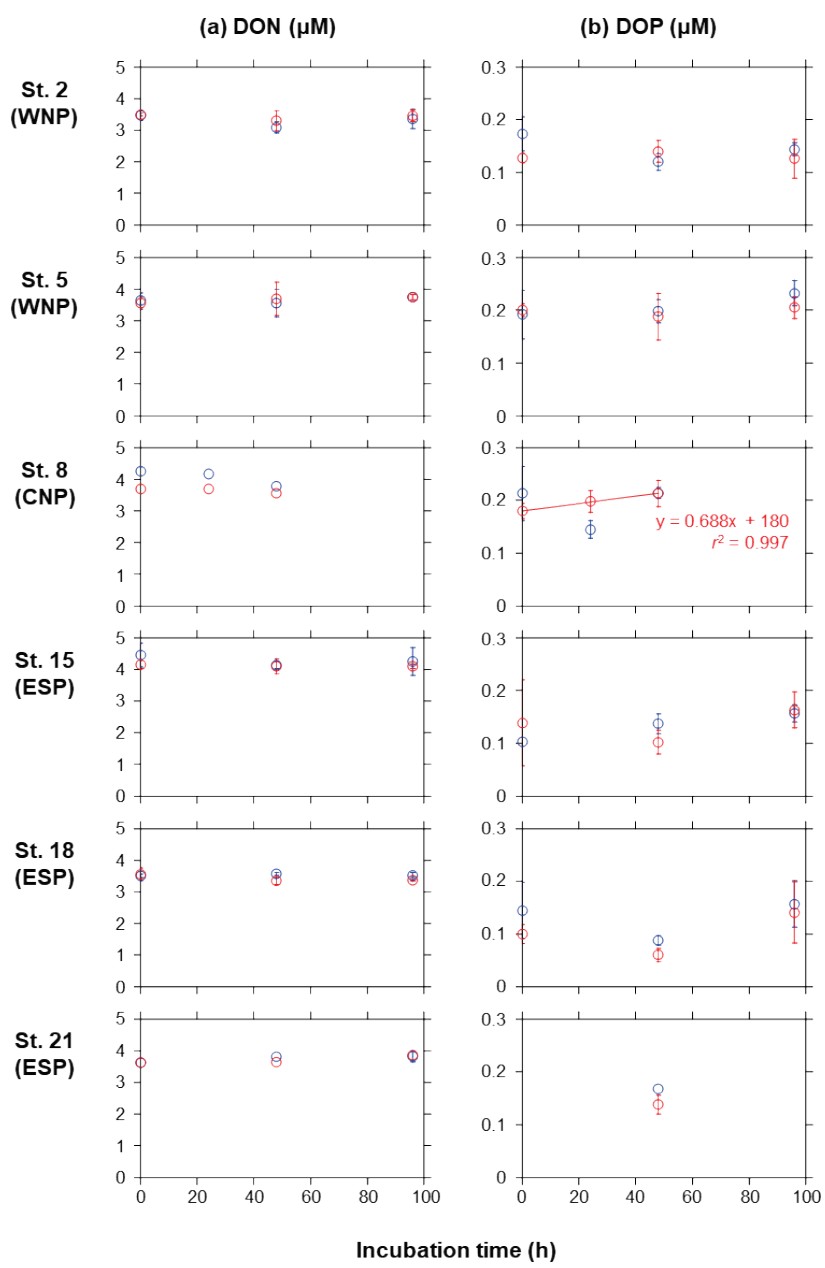


**Figure A2.** Temporal changes in concentrations of (a) DON and (b) DOP in the control (blue) and treated (red) bottles during the incubation periods at six stations during the KH-11-10 cruise. Error bars denote standard deviations. Duplicate or single data are denoted as mean or single values without error bars. A linear regression line is depicted in the DOP concentrations in the treated bottle at Station 8 as a significant increase ($p<0.05$) in the mean concentration against time was

observed.





**Figure A3.** Surface distributions of nanomolar concentrations (<1000 nM) of (a) N+N and (b) PO4 in the oligotrophic Pacific Ocean (40° N-40° S). Small black dots denote sampling stations for the nanomolar PO4 and N+N. Large orange circles denote the stations where the incubation experiments were conducted.



**Data availability.** The data are available upon request to the corresponding author (Fuminori Hashihama).

**Author contribution.** FH designed the incubation experiments. KF, HS, HO, and PWB designed the sampling schemes across the subtropical North and South Pacific. FH, TK, JK, and EMSW collected nanomolar nutrient data. STY, FH, and JK collected DON and DOP data. FH collected phytoplankton data. FH and IT performed the Fe-enrichment incubation experiment. FH wrote the manuscript. All authors reviewed and approved the manuscript.

**Competing interests.** The authors declare that they have no conflict of interest.

**Acknowledgements.** We thank the officers, crew, and scientists of the cruises of R/V Tansei Maru and R/V Hakuho Maru (Japan Agency for Marine-Earth Science and Technology) for their cooperation at sea. These cruises were performed under cooperative research system of Atmosphere and Ocean Research Institute, the University of Tokyo. We are grateful to S. Kinouchi and S. Suwa for help with sample collections during the cruises. This work was financially supported by JSPS/MEXT KAKENHI (Nos. 18067007, 22710006, 24710004, 24121001, 24121003, 24121005, 15H02802, 17H01852) and a New Zealand International visiting scientist grant.

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
