# Peer review of "Cross-basin differences in the nutrient assimilation characteristics of induced phytoplankton blooms in the subtropical Pacific waters"

_Biogeosciences, 2020_

## Referee Comment (RC1) · Anonymous Referee #1 · 14 Sep 2020

Cross-basin differences in the nutrient assimilation characteristics of induced phytoplankton blooms in the subtropical Pacific waters

by Hashihama et al. submitted to Biogeosciences

Review: The manuscript by Hashihama et al. reports on the results of a set of shipboard incubation experiments in which surface seawater collected at seven stations across the subtropical North and South Pacific ocean was amended with deep water. The additions induced phytoplankton blooms and concomitant nutrient drawdowns in all seven experiments. The novelty of the study resides mainly on the use of nutrient nanomolar measurements to reveal regional patterns in nutrient drawdown ratios. The

authors use also the obtained drawdown ratios in an attempt to contribute to current knowledge on the factors that control phosphate distribution in the subtropical Pacific Ocean.

Overall, the manuscript is very well organized and written. The introduction is fairly complete and presents nicely the context of the study. The methodology is well developed and it did not raise any major concerns from my side. The outcome is very interesting and will certainly contribute to the field of nutrient biogeochemistry in the oligotrophic ocean. The study constitutes a nice illustration on how nanomolar measurements of nutrients can give exciting insights on nutrient cycling.

I do have a few comments that may contribute to clarify some aspects of the manuscript.

General comments:

1. A recent paper published by the authors (Hashihama et al. 2020, GBC) show data from the same cruises presented in this study. If I am not wrong, this paper is referenced as Hashihama et al (submitted) at some points in the discussion section. Now that this paper has been published, and given the complementarity with this one, I strongly recommend the authors to refer to it to better put into context their outcomes. For instance, it would be interesting to check the influence of the experimental bloom-derived drawdown ratios on regional patterns of P cell quotas, addressed approximately through chlorophyll or phytoplankton biomass normalized POP data.

2. The only methodological issue that raised some concern to me was the differences in incubation times among the experiments since they can affect the drawdown ratios. This is particularly true when DIN and PO4 decreases are not linear (i.e. station 15, figure 3). I suggest the authors to conduct their calculations using only data obtained during the first 48h to see if the outcomes still stand. Otherwise, please add a statement in the discussion on the fact that different incubation times might have affected the obtained differences in nutrient drawdown ratios.

[Figure]

3. The differences among experiments in nutrient drawdowns and their ratios are sometimes small. In order to add robustness to the interpretation of these differences, the authors should calculate the errors associated to these observations (errors of the slopes of linear regressions in Figs 3a and 3b and errors of calculated $\Delta$DIN and $\Delta$PO4). More generally, I missed standard errors and/or estimated uncertainties of all data and/or calculations throughout the manuscript.

4. I would like to share with the authors some thoughts that came to my mind when reading their manuscript related to the influence of their results on P* regional distribution. I am not sure of the pertinence and accuracy of these thoughts, so I share them just in case they can feed the interpretation of their outcomes. In 4.4 section and in Figure 6, the authors make a parallelism between ambient P* and "bloom P*". This exercise is interesting and confusing at the same time. The authors state that the observed negative values of bloom P* (not observed in in situ P*) imply the presence of alternative P sources. But isn't this statement only true under the assumption of 16:1 drawdown ratios used to calculate ambient P*? Wouldn't the observed differences in ambient and bloom P* values question the use of 16:1 to estimate ambient P*? This would also be illustrated by the much higher variability in ambient P* compared to bloom P*, wouldn't it?

Specific comments: - Page 5, lines 125-126. When available, it would be useful to give the standard deviation of the triplicate analytical data. - Page 6, lines 168-169. The observed trend would be due to the low variability in DIN concentration rather than to low DIN concentration itself. - Page 11, line 219. The term 'net assimilation rates' is normally associated to tracer incorporation measurements which is not the case. Please, stick to the term drawdown rates. - Page 11, line 220. Please, add estimated errors of these slope values (cf general comment 3 above) - Page 11, lines 225-226 and page 13, line 253. I do not understand what these R2 values mean, please clarify.

---

## Referee Comment (RC2) · Anonymous Referee #2 · 27 Oct 2020

Comments to Author(s): This manuscript by Hashihama et al. describes the variation in macronutrient drawdown among Pacific Ocean surface microbial communities using deep water additions to bottle incubations. The authors present data from seven subtropical gyre sites with distinct nutrient uptake ratios, where the nutrient limiting net biological production is unknown. These observations are linked to pigment proxies for phytoplankton taxa and diatom densities to examine the role of community in nutrient drawdown ratios. The experiments yielded increased phytoplankton biomass at all sites, but varying stoichiometric ratios of uptake for DIN:PO4:Si(OH)4. I feel this manuscript strongly expands upon existing studies on the question of nitrogen, phosphorus and iron limitation in the North and South Pacific Ocean.

[Figure]

The data is presented in a straightforward manner and explained well. All sections are well written and figures are easily digested. I have a few concerns on assumptions made regarding phytoplankton pigments proxies and deep DOM composition, but otherwise recommend the manuscript should be accepted with minor revisions.

General comments

1. Regarding the methods, I request that the deep water collection be clarified slightly. It is not stated if the water was filtered to remove living cells. This is of particular concern at Stations A and 2. If unfiltered seawater was used, both grazers and microbial cells could impact the conclusions at these stations. At other stations, freezing the seawater would remove this concern, but introduce additional nutrients from burst cells. The nutrient composition of cellular detritus is likely different and more bioavailable (urea, $NH_4$, labile DOM) than deep nutrients ($NO_3$, recalcitrant DOM).

2. My largest concern is directly assuming that divinyl chlorophyll A concentrations are representative of Prochlorococcus abundances. While a useful indicator, the concentration of divinyl chlorophyll a could change between sites, season, light/depth level, etc. It is very possible Prochlorococcus cells in the Eastern South Pacific have a lower density of photosynthetic pigments, especially in the summertime at the surface where these cells were collected. In addition, since Nitrogen is limited, the cells may have adapted by lowering the concentration of N-rich photopigments further. This combined effect of photo-acclimation and adaptation to low N could explain the low divinyl chlorophyll A concentrations in the South Pacific subtropical gyre. This caveat should be acknowledged in the Discussion.

3. Regarding DOM, I had two points to consider. The authors mention more bioavailable forms of DOM that may not be present in water at 1500m. Perhaps the DOM added then would not be consumed, leading to no net changes over the incubation period. Alternatively, the balance of net uptake and release could yield no change. This possibility should be acknowledged for silicic acid as well considering the longer

incubation time and high diatom abundances at station 15.

Specific comments Line 47 - Changing 'alleviates temporarily ' to 'temporarily alleviates' reads a bit better.

Line 74 - 'observation' should be plural

Line 74-75 - I suggest changing ', to understand them, experimental validations are required.' to ', and to understand them experimental validations are required'.

Line 95 - See comment on methods

Line 120 - To what extent does the water volume in the bottle change between T0 and the last time point? Also are collection intervals than shorter for the shorter incubation times?

Line 199 - 'Dominance of Prochlorococcus' not actually quantified. See comment above.

Line 202 - A brief description for how the Nitzschia longissimi was identified should be included (i.e. by sight, microscope identification?).

Lines 312-315 (347) - See comment on phytoplankton proxies above.

Lines 333-340 - See comment on DOM above.

Line 347 - Based on uncertainty of phytoplankton composition estimates, I don't believe this phytoplankton uptake theory can be thrown out.

Line 362 - Alternatively, DOP uptake and release is balanced.

Line 396-7 - This point is very interesting and I point the authors to this short compilation reference of nitrogen fixation estimates by Bonnet et al. 2017 in (https://doi.org/10.1073/pnas.1619514114). It is possible that iron (or a trace metal) is limiting, but the microbial population does not have standing stocks of nitrogen fixers.

---

## Author Comment (AC1) · 5 Nov 2020

Reply to Referee #1 comment made on bg-2020-300, "Cross-basin differences in the nutrient assimilation characteristics of induced phytoplankton blooms in the subtropical Pacific waters " by Hashihama et al.

Review: The manuscript by Hashihama et al. reports on the results of a set of shipboard incubation experiments in which surface seawater collected at seven stations across the subtropical North and South Pacific ocean was amended with deep water. The additions induced phytoplankton blooms and concomitant nutrient drawdowns in all seven experiments. The novelty of the study resides mainly on the use of nutrient nanomolar measurements to reveal regional patterns in nutrient drawdown ratios. The authors use also the obtained drawdown ratios in an attempt to contribute to current knowledge on the factors that control phosphate distribution in the subtropical Pacific Ocean.

Overall, the manuscript is very well organized and written. The introduction is fairly complete and presents nicely the context of the study. The methodology is well developed and it did not raise any major concerns from my side. The outcome is very interesting and will certainly contribute to the field of nutrient biogeochemistry in the oligotrophic ocean. The study constitutes a nice illustration on how nanomolar measurements of nutrients can give exciting insights on nutrient cycling. I do have a few comments that may contribute to clarify some aspects of the manuscript.

*Our reply (in italics hereafter): We appreciate your constructive comments on our manuscript. We will address your comments as seen below.*

General comments:
1. A recent paper published by the authors (Hashihama et al. 2020, GBC) show data from the same cruises presented in this study. If I am not wrong, this paper is referenced as Hashihama et al (submitted) at some points in the discussion section. Now that this paper has been published, and given the complementarity with this one, I strongly recommend the authors to refer to it to better put into context their outcomes. For instance, it would be interesting to check the influence of the experimental bloom-derived drawdown ratios on regional patterns of P cell quotas, addressed approximately through chlorophyll or phytoplankton biomass normalized POP data.
*Yes, we will refer Hashihama et al. (2020, GBC) in the revised manuscript. For the suggested analysis using POP data, we have looked at the ambient data for Chl a and POP (TPP) in the study area of Hashihama et al. (2020, GBC). However, data on Chl a-normalized TPP in the mixed layer did not show a clear regional pattern (no distinctly high TPP/Chl a values in the western North Pacific). Since cell-specific content of Chl a varied depending on seasonal and/or regional light intensity, the use of Chl a-normalized TPP data is unsuitable for the suggested analysis. We consider that direct*

*observation of particulate C-N-P variability response to deep water addition will be necessary soon.*

2. The only methodological issue that raised some concern to me was the differences in incubation times among the experiments since they can affect the drawdown ratios. This is particularly true when DIN and PO4 decreases are not linear (i.e. station 15, figure 3). I suggest the authors to conduct their calculations using only data obtained during the first 48h to see if the outcomes still stand. Otherwise, please add a statement in the discussion on the fact that different incubation times might have affected the obtained differences in nutrient drawdown ratios.

*We have compared slopes of the nutrient decreases during the first 36-48 h, and for the full incubation periods (52-96 h), and both the slopes were not significantly different for DIN or PO$_4$ (paired t-test, p>0.05). We will add the related statement in the revised manuscript.*

3. The differences among experiments in nutrient drawdowns and their ratios are sometimes small. In order to add robustness to the interpretation of these differences, the authors should calculate the errors associated to these observations (errors of the slopes of linear regressions in Figs 3a and 3b and errors of calculated ΔDIN and ΔPO4). More generally, I missed standard errors and/or estimated uncertainties of all data and/or calculations throughout the manuscript.

*We will add the errors (95% confidence interval) for the slopes and intercepts of regressions and the Δ values and its ratios in the revised manuscript. The errors of the drawdown ratios can be estimated based on the error propagation rule (Miller and Miller, 1993 Statistics for Analytical Chemistry, 2$^{nd}$ edn).*

4. I would like to share with the authors some thoughts that came to my mind when reading their manuscript related to the influence of their results on P* regional distribution. I am not sure of the pertinence and accuracy of these thoughts, so I share them just in case they can feed the interpretation of their outcomes. In 4.4 section and in Figure 6, the authors make a parallelism between ambient P* and "bloom P*". This exercise is interesting and confusing at the same time. The authors state that the observed negative values of bloom P* (not observed in in situ P*) imply the presence of alternative P sources. But isn't this statement only true under the assumption of 16:1 drawdown ratios used to calculate ambient P*? Wouldn't the observed differences in ambient and bloom P* values question the use of 16:1 to estimate ambient P*? This would also be illustrated by the much higher variability in ambient P* compared to bloom P*, wouldn't it?

*Apologies, we had stated a slightly confusing discussion in 4.4 section. In the revised manuscript, we will state a simple discussion as "The difference between the bloom P* and ambient P* largely depends on the different N:P consumption ratios of <16 and 16, respectively. If the low N:P consumption ratios (<16) are consistently dominant in the PO$_4$-depleted WNP, alternative P sources*

*other than PO$_4$ are required to fully exhaust DIN. Since lower DOP concentrations and higher alkaline phosphatase activity were observed in the WNP, compared to other subtropical Pacific regions (Hashihama et al., 2019; Hashihama et al., 2020), active DOP utilization in the WNP likely contributes to the DIN exhaustion. These perspectives suggest that, in the studies on subtropical nutrient biogeochemistry using N:P stoichiometry, the bioavailable fraction of DOP could be an important factor as well as DIN and PO$_4$."*

Specific comments:

- Page 5, lines 125-126. When available, it would be useful to give the standard deviation of the triplicate analytical data.

*The standard deviations will be indicated in Table 2.*

- Page 6, lines 168-169. The observed trend would be due to the low variability in DIN concentration rather than to low DIN concentration itself.

*We will add "the consistently low concentrations of DIN" in the revised manuscript.*

- Page 11, line 219. The term 'net assimilation rates' is normally associated to tracer incorporation measurements which is not the case. Please, stick to the term drawdown rates.

*We will use "drawdown rates" throughout the text.*

- Page 11, line 220. Please, add estimated errors of these slope values (cf general comment 3 above)

*Yes, we will do as mentioned above.*

- Page 11, lines 225-226 and page 13, line 253. I do not understand what these R2 values mean, please clarify.

*These values were derived from the linear regression analysis. We will add "$r^2$=xxx in linear regression" in these parts.*

---

## Author Comment (AC2) · 5 Nov 2020

Reply to Referee #2 comment made on bg-2020-300, "Cross-basin differences in the nutrient assimilation characteristics of induced phytoplankton blooms in the subtropical Pacific waters
" by Hashihama et al.

Comments to Author(s): This manuscript by Hashihama et al. describes the variation in macronutrient drawdown among Pacific Ocean surface microbial communities using deep water additions to bottle incubations. The authors present data from seven subtropical gyre sites with distinct nutrient uptake ratios, where the nutrient limiting net biological production is unknown. These observations are linked to pigment proxies for phytoplankton taxa and diatom densities to examine the role of community in nutrient drawdown ratios. The experiments yielded increased phytoplankton biomass at all sites, but varying stoichiometric ratios of uptake for DIN:PO4:Si(OH)4. I feel this manuscript strongly expands upon existing studies on the question of nitrogen, phosphorus and iron limitation in the North and South Pacific Ocean.

The data is presented in a straightforward manner and explained well. All sections are well written and figures are easily digested. I have a few concerns on assumptions made regarding phytoplankton pigments proxies and deep DOM composition, but otherwise recommend the manuscript should be accepted with minor revisions.

*Our reply (in italics hereafter): Thank you for your constructive comments. We will address your comments as seen below.*

General comments
1. Regarding the methods, I request that the deep water collection be clarified slightly. It is not stated if the water was filtered to remove living cells. This is of particular concern at Stations A and 2. If unfiltered seawater was used, both grazers and microbial cells could impact the conclusions at these stations. At other stations, freezing the seawater would remove this concern, but introduce additional nutrients from burst cells. The nutrient composition of cellular detritus is likely different and more bioavailable (urea, NH4, labile DOM) than deep nutrients (NO3, recalcitrant DOM).

*We used the unfiltered deep water from 1500 m depth to avoid ammonium contamination from filtration process. As you have considered, there might be a potential influence of heterotrophs in the deep water on the results of incubation experiments. However, concentration of particulate organic carbon in the deep layer below 1000 m is generally less than 10% of that in the surface layer (Hebel and Karl, 2001 DSR-II 48, 1669-1695; Yamada et al., 2017 MEPS 583, 81-93) and prokaryotic abundance and production exponentially decrease with depth (Yokokawa et al., 2013 LO 58, 61-73). Furthermore, the proportion of deep water to total incubated volume (surface water + deep water) was only 2.1% as*

*stated in the section 2.2. The large dilution was also confirmed from T0 data of DON and DOP in the treated bottles which were not significantly different from those in the control bottles (paired t-test, p>0.05, Fig. A2), indicating that the influence of labile DON and DOP additions were negligible. Thus, we conclude that the influences of heterotrophs and labile DOM supply by freezing were at negligible levels compared to the large enrichments of inorganic nutrients.*

2. My largest concern is directly assuming that divinyl chlorophyll A concentrations are representative of Prochlorococcus abundances. While a useful indicator, the concentration of divinyl chlorophyll a could change between sites, season, light/depth level, etc. It is very possible Prochlorococcus cells in the Eastern South Pacific have a lower density of photosynthetic pigments, especially in the summertime at the surface where these cells were collected. In addition, since Nitrogen is limited, the cells may have adapted by lowering the concentration of N-rich photopigments further. This combined effect of photo-acclimation and adaptation to low N could explain the low divinyl chlorophyll A concentrations in the South Pacific subtropical gyre. This caveat should be acknowledged in the Discussion.

*We agree that seasonal/regional PAR level influences the cellular pigment quotas. In contrast, N limitation occurred not only in the South Pacific but also in the North Pacific, because ambient $DIN:PO_4$ ratios (≤8) were much lower than the Redfield ratio (16) and subtropical particulate N:P ratio (28) (Martiny et al., 2013 Nat. Geosci. 6, 279-283). Therefore, N limitation might not be a robust reason for any seasonal/regional variations in the pigment concentrations, including DVchl a. Furthermore, we consider that pigment ratios (Tfuco:Zea (no N in either of the pigments) and DVchl a:Tchl a (both pigments contain N)) are useful for comparing the regional variations in phytoplankton composition even if seasonal/regional differences of photo-acclimation/adaptation and N-limitation occur. In the perspective of phytoplankton physiology, Tfuco and Zea play roles in light-harvesting and photoprotection, respectively (Falkowski, 2013 Aquatic Photosynthesis 2nd edn). Based on these roles, Zea content in cyanobacteria should be higher in the high-PAR South Pacific than the low-PAR North Pacific, while Tfuco content in eukaryotes should be higher in the low-PAR North Pacific than the high-PAR South Pacific. However, ambient Zea (Tfuco) concentrations were higher (lower) in the North Pacific than the South Pacific, indicating that the biomass proportion of cyanobacteria to eukaryotes was higher in the North Pacific than the South Pacific. Thus, although we will revise the statements for seasonal/regional variation in each pigment concentration, we do not revise the statements for pigment ratios and seasonal/regional variations in phytoplankton composition.*

3. Regarding DOM, I had two points to consider. The authors mention more bioavailable forms of DOM that may not be present in water at 1500m. Perhaps the DOM added then would not be consumed, leading to no net changes over the incubation period. Alternatively, the balance of net uptake and

release could yield no change. This possibility should be acknowledged for silicic acid as well considering the longer incubation time and high diatom abundances at station 15.

*We also interpret that DON and DOP in the deep water did not have an influence on various parameters in the incubation, because these additions were negligible due to the large dilution as mentioned above. Since the resident phytoplankton in the DIN and PO₄-depleted subtropical ocean might have a high affinity to DON and DOP, rather than DIN and PO₄, we considered the possibility that phytoplankton in the incubated bottles consume labile DON and DOP in the surface water (not in the deep water) rather than DIN and PO₄. However, such a phenomenon did not occur in the present study and the uptake and release of DON and DOP were balanced. As you suggest, the uptake and release of silicic acid likely occurred. We will add the related statement in the revised manuscript.*

Specific comments

Line 47 - Changing 'alleviates temporarily ' to 'temporarily alleviates' reads a bit better.

*We will revise as you have suggested.*

Line 74 - 'observation' should be plural

*Yes, it will be plural.*

Line 74-75 - I suggest changing ', to understand them, experimental validations are required.' to ', and to understand them experimental validations are required'.

*We will revise based on your suggestion.*

Line 95 - See comment on methods

*See our answer above.*

Line 120 - To what extent does the water volume in the bottle change between T0 and the last time point? Also are collection intervals than shorter for the shorter incubation times?

*At the end point, water volumes in the bottles just prior to final sampling were approximately 1.8 L, because the subsamples for nanomolar nutrients and DON/P were collected from the initial volume of 2.35 L (surface water + deep water). Also, the collection intervals were shorter for the shorter incubation times. We subsampled 5-6 times for nanomolar nutrients and 3 times for DON and DOP during the incubation periods for 48-96 h. We will add these statements in the revised manuscript.*

Line 199 - 'Dominance of Prochlorococcus' not actually quantified. See comment above.

*See our answer above.*

Line 202 - A brief description for how the Nitzschia longissimi was identified should be included (i.e. by sight, microscope identification?).

*We described the microscopic analysis of diatoms in the section 2.6. of Materials and methods.*

Lines 312-315 (347) - See comment on phytoplankton proxies above.

*See our answer above.*

Lines 333-340 - See comment on DOM above.

*See our answer above.*

Line 347 - Based on uncertainty of phytoplankton composition estimates, I don't believe this phytoplankton uptake theory can be thrown out.

*See our answer above.*

Line 362 - Alternatively, DOP uptake and release is balanced.

*We will revise based on your suggestion.*

Line 396-7 - This point is very interesting and I point the authors to this short compilation reference of nitrogen fixation estimates by Bonnet et al. 2017 in (https://doi.org/10.1073/pnas.1619514114). It is possible that iron (or a trace metal) is limiting, but the microbial population does not have standing stocks of nitrogen fixers.

*We will refer Bonnet et al. (2017 PNAS) in the revised manuscript.*

---

## Author Response (AR1)

Reply to Referee #1 comment made on bg-2020-300, "Cross-basin differences in the nutrient assimilation characteristics of induced phytoplankton blooms in the subtropical Pacific waters
" by Hashihama et al.

Review: The manuscript by Hashihama et al. reports on the results of a set of shipboard incubation experiments in which surface seawater collected at seven stations across the subtropical North and South Pacific ocean was amended with deep water. The additions induced phytoplankton blooms and concomitant nutrient drawdowns in all seven experiments. The novelty of the study resides mainly on the use of nutrient nanomolar measurements to reveal regional patterns in nutrient drawdown ratios. The authors use also the obtained drawdown ratios in an attempt to contribute to current knowledge on the factors that control phosphate distribution in the subtropical Pacific Ocean.

Overall, the manuscript is very well organized and written. The introduction is fairly complete and presents nicely the context of the study. The methodology is well developed and it did not raise any major concerns from my side. The outcome is very interesting and will certainly contribute to the field of nutrient biogeochemistry in the oligotrophic ocean. The study constitutes a nice illustration on how nanomolar measurements of nutrients can give exciting insights on nutrient cycling. I do have a few comments that may contribute to clarify some aspects of the manuscript.

*Our reply (in italics hereafter): We appreciate your constructive comments on our manuscript. We have addressed your comments as seen below.*

General comments:

1. A recent paper published by the authors (Hashihama et al. 2020, GBC) show data from the same cruises presented in this study. If I am not wrong, this paper is referenced as Hashihama et al (submitted) at some points in the discussion section. Now that this paper has been published, and given the complementarity with this one, I strongly recommend the authors to refer to it to better put into context their outcomes. For instance, it would be interesting to check the influence of the experimental bloom-derived drawdown ratios on regional patterns of P cell quotas, addressed approximately through chlorophyll or phytoplankton biomass normalized POP data.

*Yes, we have referred Hashihama et al. (2020, GBC) in the revised manuscript. For the suggested analysis using POP data, we have looked at the ambient data for Chl a and POP (TPP) in the study area of Hashihama et al. (2020, GBC). However, data on Chl a-normalized TPP in the mixed layer did not show a clear regional pattern (no distinctly high TPP/Chl a values in the western North Pacific). Since cell-specific content of Chl a varies depending on seasonal and/or regional light intensity, the use of Chl a-normalized TPP data is unsuitable for the suggested analysis. We consider*

*that direct observation of particulate C-N-P variability response to deep water addition will be necessary soon.*

2. The only methodological issue that raised some concern to me was the differences in incubation times among the experiments since they can affect the drawdown ratios. This is particularly true when DIN and PO4 decreases are not linear (i.e. station 15, figure 3). I suggest the authors to conduct their calculations using only data obtained during the first 48h to see if the outcomes still stand. Otherwise, please add a statement in the discussion on the fact that different incubation times might have affected the obtained differences in nutrient drawdown ratios.

*We have compared slopes of the nutrient decreases during the first 36-48 h, and for the full incubation periods (52-96 h), and both the slopes were not significantly different for DIN or $PO_4$ (paired t-test, p>0.05). We have added the related statement in the revised manuscript (L223-224).*

3. The differences among experiments in nutrient drawdowns and their ratios are sometimes small. In order to add robustness to the interpretation of these differences, the authors should calculate the errors associated to these observations (errors of the slopes of linear regressions in Figs 3a and 3b and errors of calculated ΔDIN and ΔPO4). More generally, I missed standard errors and/or estimated uncertainties of all data and/or calculations throughout the manuscript.

*We have added the errors (95% confidence interval) for the slopes and intercepts of regressions and the Δ values and its ratios in the revised manuscript. The errors of the drawdown ratios were calculated based on the error propagation rule (Miller and Miller, 1993 Statistics for Analytical Chemistry, 2nd edn) as stated in L121-122. In this calculation, the mean drawdown ratios changed slightly from those in the previous manuscript due to the use of different significant digits, but it did not affect the conclusion.*

4. I would like to share with the authors some thoughts that came to my mind when reading their manuscript related to the influence of their results on P* regional distribution. I am not sure of the pertinence and accuracy of these thoughts, so I share them just in case they can feed the interpretation of their outcomes. In 4.4 section and in Figure 6, the authors make a parallelism between ambient P* and "bloom P*". This exercise is interesting and confusing at the same time. The authors state that the observed negative values of bloom P* (not observed in in situ P*) imply the presence of alternative P sources. But isn't this statement only true under the assumption of 16:1 drawdown ratios used to calculate ambient P*? Wouldn't the observed differences in ambient and bloom P* values question the use of 16:1 to estimate ambient P*? This would also be illustrated by the much higher variability in ambient P* compared to bloom P*, wouldn't it?

*Apologies, we had stated a slightly confusing discussion in 4.4 section. In the revised manuscript, we*

*have stated a simple discussion as "The difference between the bloom P\* and ambient P\* largely depends on the different N:P consumption ratios of ≤13.3 and 16, respectively. If the low N:P consumption ratios (≤13.3) are consistently dominant in the $PO_4$-depleted WNP, alternative P sources other than $PO_4$ would be required to fully exhaust DIN. Since lower DOP concentrations and higher alkaline phosphatase activity were observed in the WNP, compared to other subtropical Pacific regions (Hashihama et al., 2019; Hashihama et al., 2020), active DOP utilization in the WNP likely contributes to the DIN exhaustion. These perspectives suggest that, in the studies on subtropical nutrient biogeochemistry using N:P stoichiometry, the bioavailable fraction of DOP could be an important factor as well as DIN and $PO_4$." (L413-420)*

Specific comments:

- Page 5, lines 125-126. When available, it would be useful to give the standard deviation of the triplicate analytical data.

*The standard deviations have indicated in Table 2.*

- Page 6, lines 168-169. The observed trend would be due to the low variability in DIN concentration rather than to low DIN concentration itself.

*We have added "the consistently low concentrations of DIN" in the revised manuscript (L172).*

- Page 11, line 219. The term 'net assimilation rates' is normally associated to tracer incorporation measurements which is not the case. Please, stick to the term drawdown rates.

*We have used "drawdown rates" throughout the text.*

- Page 11, line 220. Please, add estimated errors of these slope values (cf general comment 3 above)

*Yes, we have added as mentioned above.*

- Page 11, lines 225-226 and page 13, line 253. I do not understand what these R2 values mean, please clarify.

*These values were derived from the linear regression analysis. We have added "$r^2$=xxx in linear regression" in these parts (L230; L259).*

Reply to Referee #2 comment made on bg-2020-300, "Cross-basin differences in the nutrient assimilation characteristics of induced phytoplankton blooms in the subtropical Pacific waters
" by Hashihama et al.

Comments to Author(s): This manuscript by Hashihama et al. describes the variation in macronutrient drawdown among Pacific Ocean surface microbial communities using deep water additions to bottle incubations. The authors present data from seven subtropical gyre sites with distinct nutrient uptake ratios, where the nutrient limiting net biological production is unknown. These observations are linked to pigment proxies for phytoplankton taxa and diatom densities to examine the role of community in nutrient drawdown ratios. The experiments yielded increased phytoplankton biomass at all sites, but varying stoichiometric ratios of uptake for DIN:PO4:Si(OH)4. I feel this manuscript strongly expands upon existing studies on the question of nitrogen, phosphorus and iron limitation in the North and South Pacific Ocean.

The data is presented in a straightforward manner and explained well. All sections are well written and figures are easily digested. I have a few concerns on assumptions made regarding phytoplankton pigments proxies and deep DOM composition, but otherwise recommend the manuscript should be accepted with minor revisions.

*Our reply (in italics hereafter): Thank you for your constructive comments. We have addressed your comments as seen below.*

General comments
1. Regarding the methods, I request that the deep water collection be clarified slightly. It is not stated if the water was filtered to remove living cells. This is of particular concern at Stations A and 2. If unfiltered seawater was used, both grazers and microbial cells could impact the conclusions at these stations. At other stations, freezing the seawater would remove this concern, but introduce additional nutrients from burst cells. The nutrient composition of cellular detritus is likely different and more bioavailable (urea, NH4, labile DOM) than deep nutrients (NO3, recalcitrant DOM).

*We used the unfiltered deep water from 1500 m depth to avoid ammonium contamination from filtration process. This has been stated in the revised manuscript (L105-106). As you have considered, there might be a potential influence of heterotrophs in the deep water on the results of incubation experiments. However, concentration of particulate organic carbon in the deep layer below 1000 m is generally less than 10% of that in the surface layer (Hebel and Karl, 2001 DSR-II 48, 1669-1695; Yamada et al., 2017 MEPS 583, 81-93) and prokaryotic abundance and production exponentially decrease with depth (Yokokawa et al., 2013 LO 58, 61-73). Furthermore, the proportion of deep water*

*to the total incubated volume (surface water + deep water) was only 2.1% as stated in the section 2.2. The large dilution was also confirmed from T0 data of DON and DOP in the treated bottles which were not significantly different from those in the control bottles (paired t-test, p>0.05, Fig. A2), indicating that the influence of labile DON and DOP additions were negligible. Thus, we conclude that the influences of heterotrophs and labile DOM supply by freezing were at negligible levels compared to the large enrichments of inorganic nutrients.*

2. My largest concern is directly assuming that divinyl chlorophyll A concentrations are representative of Prochlorococcus abundances. While a useful indicator, the concentration of divinyl chlorophyll a could change between sites, season, light/depth level, etc. It is very possible Prochlorococcus cells in the Eastern South Pacific have a lower density of photosynthetic pigments, especially in the summertime at the surface where these cells were collected. In addition, since Nitrogen is limited, the cells may have adapted by lowering the concentration of N-rich photopigments further. This combined effect of photo-acclimation and adaptation to low N could explain the low divinyl chlorophyll A concentrations in the South Pacific subtropical gyre. This caveat should be acknowledged in the Discussion.

*We agree that seasonal/regional PAR level influences the cellular pigment quotas. In contrast, N limitation occurred not only in the South Pacific but also in the North Pacific, because ambient DIN:PO$_4$ ratios (≤8) were much lower than the Redfield ratio (16) and subtropical particulate N:P ratio (28) (Martiny et al., 2013 Nat. Geosci. 6, 279-283). Therefore, N limitation might not be a robust reason for any seasonal/regional variations in the pigment concentrations, including DVchl a. Furthermore, we consider that pigment ratios (Tfuco:Zea (no N in either of the pigments) and DVchl a:Tchl a (both pigments contain N)) are useful for comparing the regional variations in phytoplankton composition even if seasonal/regional differences of photo-acclimation/adaptation and N-limitation occur. In the perspective of phytoplankton physiology, Tfuco and Zea play roles in light-harvesting and photoprotection, respectively (Falkowski, 2013 Aquatic Photosynthesis 2$^{nd}$ edn). Based on these roles, Zea content in cyanobacteria should be higher in the high-PAR South Pacific than the low-PAR North Pacific, while Tfuco content in eukaryotes should be higher in the low-PAR North Pacific than the high-PAR South Pacific. However, ambient Zea (Tfuco) concentrations were higher (lower) in the North Pacific than the South Pacific, indicating that the biomass proportion of cyanobacteria to eukaryotes was higher in the North Pacific than the South Pacific. Thus, we did not revise the statements for pigment ratios and seasonal/regional variations in phytoplankton composition.*

3. Regarding DOM, I had two points to consider. The authors mention more bioavailable forms of DOM that may not be present in water at 1500m. Perhaps the DOM added then would not be consumed, leading to no net changes over the incubation period. Alternatively, the balance of net uptake and

release could yield no change. This possibility should be acknowledged for silicic acid as well considering the longer incubation time and high diatom abundances at station 15.

*We also interpret that DON and DOP in the deep water did not have an influence on various parameters in the incubation, because these additions were negligible due to the large dilution as mentioned above. Since the resident phytoplankton in the DIN and PO₄-depleted subtropical ocean might have a high affinity to DON and DOP, rather than DIN and PO₄, we considered the possibility that phytoplankton in the incubated bottles consume labile DON and DOP in the surface water (not in the deep water) rather than DIN and PO₄. However, such a phenomenon did not occur in the present study and the uptake and release of DON and DOP were balanced. As you suggest, the uptake and release of silicic acid likely occurred. We have added the related statement in the revised manuscript (L380-381).*

Specific comments

Line 47 - Changing 'alleviates temporarily ' to 'temporarily alleviates' reads a bit better.

*We have revised as you have suggested (L47).*

Line 74 - 'observation' should be plural

*Yes, it has been plural (L74).*

Line 74-75 - I suggest changing ', to understand them, experimental validations are required.' to ', and to understand them experimental validations are required'.

*We have revised based on your suggestion (L74-75).*

Line 95 - See comment on methods

*See our answer above.*

Line 120 - To what extent does the water volume in the bottle change between T0 and the last time point? Also are collection intervals than shorter for the shorter incubation times?

*At the end point, water volumes in the bottles just prior to final sampling were approximately 1.8 L, because the subsamples for nanomolar nutrients and DON/P were collected from the initial volume of 2.35 L (surface water + deep water). Also, the collection intervals were shorter for the shorter incubation times. We subsampled 5-6 times for nanomolar nutrients and 3 times for DON and DOP during the incubation periods for 48-96 h. We have added these statements in the revised manuscript (L114-117).*

Line 199 - 'Dominance of Prochlorococcus' not actually quantified. See comment above.

*We deleted this statement.*

Line 202 - A brief description for how the Nitzschia longissimi was identified should be included (i.e. by sight, microscope identification?).

*We described the microscopic analysis of diatoms in the section 2.6. of Materials and methods (L147-150).*

Lines 312-315 (347) - See comment on phytoplankton proxies above.

*See our answer above.*

Lines 333-340 - See comment on DOM above.

*See our answer above.*

Line 347 - Based on uncertainty of phytoplankton composition estimates, I don't believe this phytoplankton uptake theory can be thrown out.

*See our answer above.*

Line 362 - Alternatively, DOP uptake and release is balanced.

*We have revised based on your suggestion (L369-370).*

Line 396-7 - This point is very interesting and I point the authors to this short compilation reference of nitrogen fixation estimates by Bonnet et al. 2017 in (https://doi.org/10.1073/pnas.1619514114). It is possible that iron (or a trace metal) is limiting, but the microbial population does not have standing stocks of nitrogen fixers.

*We have referred Bonnet et al. (2017 PNAS) in the revised manuscript (L408).*

[revised manuscript text omitted]

**St. A (WNP)**
$y = -6.46x + 769$
$r^2 = 0.886$
$y = 2.92x - 15.4$
$r^2 = 0.903$

**St. 2 (WNP)**
$y = -1.39x + 837$
$r^2 = 0.950$
$y = 0.0792x + 1.60$
$r^2 = 0.970$

**St. 5 (WNP)**
$y = -1.80x + 795$
$r^2 = 0.864$
$y = 0.0917x + 2.60$
$r^2 = 0.781$

**St. 8 (CNP)**
$y = -3.53x + 889$
$r^2 = 0.792$
$y = 0.225x + 2.00$
$r^2 = 0.782$
$y = 0.183x + 1.40$
$r^2 = 0.953$

**St. 15 (ESP)**
$y = -7.17x + 963$
$r^2 = 0.856$
$y = 0.117x - 1.20$
$r^2 = 0.841$
$y = -0.713x + 297$
$r^2 = 0.919$

**St. 18 (ESP)**
$y = -2.64x + 868$
$r^2 = 0.926$
$y = 0.0792x + 1.40$
$r^2 = 0.930$
$y = 0.0333x + 0.600$
$r^2 = 0.941$

**St. 21 (ESP)**
$y = -2.62x + 832$
$r^2 = 0.993$
$y = 0.0833x + 3.60$
$r^2 = 0.971$

**Incubation time (h)**

[Figure]

**Figure A1.** Temporal changes in concentrations of (a) NO₃, (b) NO₂, and (c) NH₄ in the control (blue) and treated (red) bottles during the incubation periods at seven stations. Error bars denote standard deviations ($n$=3). Duplicate or single data are denoted as mean or single values without error bars. Linear regression lines are depicted when significant decreases or increases ($p$<0.05) in the mean concentrations against time were observed. Errors of slope and intercept represent 95% confidence intervals.

書式を変更: フォント：斜体

[Figure]

**(a) DON (µM)**      **(b) DOP (µM)**

St. 2 (WNP)
St. 5 (WNP)
St. 8 (CNP)
St. 15 (ESP)
St. 18 (ESP)
St. 21 (ESP)

$y = 0.688x + 180$
$r^2 = 0.997$

**Incubation time (h)**

[Figure]

**(a) DON (µM)** **(b) DOP (µM)**

Slope = 0.69 ± 0.07
Intercept = 181 ± 2
$r^2$ = 0.997

**Incubation time (h)**

[revised manuscript text omitted]

書式を変更: フォント：(日) MS 明朝, (言語 1) 日本語

---

## Author Response (AR3)

Reply to Associate Editor comment made on bg-2020-300, "Cross-basin differences in the nutrient assimilation characteristics of induced phytoplankton blooms in the subtropical Pacific waters
" by Hashihama et al.

General comments 1 and 2 raised by Referee #2 are addressed in your reply-to-referee document, yet they are not accounted for your revised manuscript. Although these comments do not question your conclusions, I believe that they should be discussed in the manuscript as other readers may have the same concerns as Referee #2.

*Our reply (in italics hereafter): We appreciate your constructive comments. In the uploaded revised manuscript, we have added the statements associated with the general comments 1 (L105-109; L282-284) and 2 (L179-189) from Referee #2.*

[revised manuscript text omitted]